# Few for Many: Tchebycheff Set Scalarization for Many-Objective Optimization

**Xi Lin[1], Yilu Liu[1], Xiaoyuan Zhang[1], Fei Liu[1], Zhenkun Wang[2], Qingfu Zhang[1]***
[1]City University of Hong Kong, [2]Southern University of Science and Technology
`{xi.lin, yiluliu3, xzhang2523-c, fliu36-c}@my.cityu.edu.hk`
`wangzhenkun90@gmail.com, qingfu.zhang@cityu.edu.hk`

## Abstract

Multi-objective optimization can be found in many real-world applications where some conflicting objectives can not be optimized by a single solution. Existing optimization methods often focus on finding a set of Pareto solutions with different optimal trade-offs among the objectives. However, the required number of solutions to well approximate the whole Pareto optimal set could be exponentially large with respect to the number of objectives, which makes these methods unsuitable for handling many optimization objectives. In this work, instead of finding a dense set of Pareto solutions, we propose a novel Tchebycheff set scalarization method to find a few representative solutions (e.g., 5) to cover a large number of objectives (e.g., $> 100$) in a collaborative and complementary manner. In this way, each objective can be well addressed by at least one solution in the small solution set. In addition, we further develop a smooth Tchebycheff set scalarization approach for efficient optimization with good theoretical guarantees. Experimental studies on different problems with many optimization objectives demonstrate the effectiveness of our proposed method.

## 1 Introduction

In real-world applications, it is very often that many optimization objectives should be considered at the same time. Examples include manufacturing or engineering design with various specifications to achieve (Adriana et al., 2018; Wang et al., 2020), decision-making systems with different factors to consider (Roijers et al., 2014; Hayes et al., 2022), and molecular generation with multiple criteria to satisfy (Jain et al., 2023; Zhu et al., 2023). For a non-trivial problem, these optimization objectives conflict one another. Therefore, it is very difficult, if not impossible, for a single solution to accommodate all objectives at the same time (Miettinen, 1999; Ehrgott, 2005).

In the past several decades, much effort has been made to develop efficient algorithms for finding a set of Pareto solutions with diverse optimal trade-offs among different objectives. However, the Pareto set that contains all optimal trade-off solutions could be an manifold in the decision space, of which the dimensionality can be large for a problem with many objectives (Hillermeier, 2001). The number of required solutions to well approximate the whole Pareto set will increase exponentially with the number of objectives, which leads to prohibitively high computational overhead. In addition, a large solution set with high-dimensional objective vectors could easily become unmanageable for decision-makers. Indeed, a problem with more than 3 objectives is already called the many-objective optimization problem (Fleming et al., 2005; Ishibuchi et al., 2008), and existing methods will struggle to deal with problems with a significantly larger number of optimization objectives (Sato & Ishibuchi, 2023).

In this work, instead of finding a dense set of Pareto solutions, we investigate a new approach for many-objective optimization, which aims to find a small set of solutions (e.g., 5) to handle a large number of objectives (e.g., $> 100$). In the optimal case, each objective should be well addressed by at least one solution in the small solution set as illustrated in Figure 1(d). This setting is important for different real-world applications with many objectives to optimize, such as finding complementary

---
*Corresponding author.

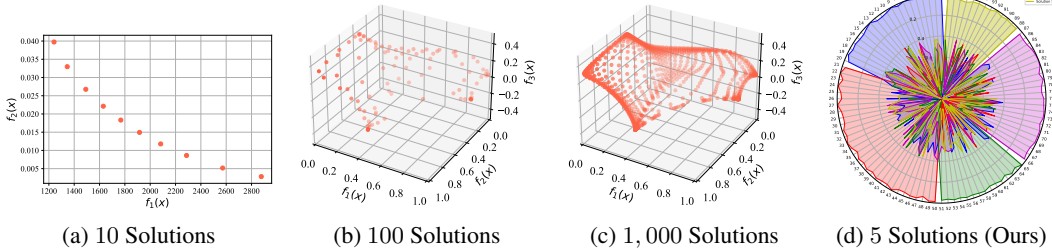

| (a) 10 Solutions | (b) 100 Solutions | (c) 1,000 Solutions | (d) 5 Solutions (Ours) |

Figure 1: **Large Set v.s. Small Set for Multi-Objective Optimization. (a)(b)(c) Large Set:** Classic algorithms use 10, 100 and 1000 solutions to approximate the whole Pareto front for 2 and 3-objective optimization problems. The required number of solutions for a good approximation could increase exponentially with the number of objectives. **(d) Small Set:** This work investigates how to efficiently find a few solutions (e.g., 5) to collaboratively handle many optimization objectives (e.g., 100).

engineering designs to satisfy various criteria (Fleming et al., 2005), producing a few different versions of advertisements to serve a large group of diverse audiences (Matz et al., 2017; Eckles et al., 2018), and building a small set of models to handle many different data (Yi et al., 2014; Zhong et al., 2016) or tasks (Standley et al., 2020; Fifty et al., 2021). However, this demand has received little attention from the multi-objective optimization community. To properly handle this setting, this work makes the following contributions [1]:

- We propose a novel Tchebycheff set (TCH-Set) scalarization approach to find a few optimal solutions in a collaborative and complementary manner for many-objective optimization.

- We further develop a smooth Tchebycheff set (STCH-Set) scalarization approach to tackle the non-smoothness of TCH-Set scalarization for efficient gradient-based optimization.

- We provide theoretical analyses to show that our proposed approaches enjoy good theoretical properties for multi-objective optimization.

- We conduct experiments on various multi-objective optimization problems with many objectives to demonstrate the efficiency of our proposed method.

## 2 PRELIMINARIES AND RELATED WORK

### 2.1 MULTI-OBJECTIVE OPTIMIZATION

In this work, we consider the following multi-objective optimization problem:

$$\min_{\boldsymbol{x} \in \mathcal{X}} \boldsymbol{f}(\boldsymbol{x}) = (f_1(\boldsymbol{x}), f_2(\boldsymbol{x}), \cdots, f_m(\boldsymbol{x})), \tag{1}$$

where $\boldsymbol{x} \in \mathcal{X}$ is a solution and $\boldsymbol{f}(\boldsymbol{x}) = (f_1(\boldsymbol{x}), f_2(\boldsymbol{x}), \cdots, f_m(\boldsymbol{x})) \in \mathbb{R}^m$ are $m$ differentiable objective functions. For a non-trivial problem, there is no single solution $\boldsymbol{x}^*$ that can optimize all objective functions at the same time. Therefore, we have the following definitions of dominance, (weakly) Pareto optimality, and Pareto set/front for multi-objective optimization (Miettinen, 1999):

**Definition 1** (Dominance and Strict Dominance). *Let $\boldsymbol{x}^{(a)}, \boldsymbol{x}^{(b)} \in \mathcal{X}$ be two solutions for problem (1), $\boldsymbol{x}^{(a)}$ is said to dominate $\boldsymbol{x}^{(b)}$, denoted as $\boldsymbol{f}(\boldsymbol{x}^{(a)}) \prec \boldsymbol{f}(\boldsymbol{x}^{(b)})$, if and only if $f_i(\boldsymbol{x}^{(a)}) \leq f_i(\boldsymbol{x}^{(b)}) \,\forall i \in \{1, ..., m\}$ and $f_j(\boldsymbol{x}^{(a)}) < f_j(\boldsymbol{x}^{(b)}) \,\exists j \in \{1, ..., m\}$. In addition, $\boldsymbol{x}^{(a)}$ is said to strictly dominate $\boldsymbol{x}^{(b)}$ (i.e., $\boldsymbol{f}(\boldsymbol{x}^{(a)}) \prec_{strict} \boldsymbol{f}(\boldsymbol{x}^{(b)})$), if and only if $f_i(\boldsymbol{x}^{(a)}) < f_i(\boldsymbol{x}^{(b)}) \,\forall i \in \{1, ..., m\}$.*

**Definition 2** ((Weakly) Pareto Optimality). *A solution $\boldsymbol{x}^* \in \mathcal{X}$ is Pareto optimal if there is no $\boldsymbol{x} \in \mathcal{X}$ such that $\boldsymbol{f}(\boldsymbol{x}) \prec \boldsymbol{f}(\boldsymbol{x}^*)$. A solution $\boldsymbol{x}' \in \mathcal{X}$ is weakly Pareto optimal if there is no $\boldsymbol{x} \in \mathcal{X}$ such that $\boldsymbol{f}(\boldsymbol{x}) \prec_{strict} \boldsymbol{f}(\boldsymbol{x}')$.*

**Definition 3** (Pareto Set and Pareto Front). *The set of all Pareto optimal solutions $\boldsymbol{X}^* = \{\boldsymbol{x} \in \mathcal{X} | \boldsymbol{f}(\hat{\boldsymbol{x}}) \not\prec \boldsymbol{f}(\boldsymbol{x}) \,\forall \hat{\boldsymbol{x}} \in \mathcal{X}\}$ is called the Pareto set. Its image in the objective space $\boldsymbol{f}(\boldsymbol{X}^*) = \{\boldsymbol{f}(\boldsymbol{x}) \in \mathbb{R}^m | \boldsymbol{x} \in \boldsymbol{X}^*\}$ is called the Pareto front.*

---

[1]Our source code is available at: https://github.com/Xi-L/STCH-Set

Under mild conditions, the Pareto set and front could be on an $(m-1)$-dimensional manifold in the decision or objective space (Hillermeier, 2001), which contains infinite Pareto solutions. Many optimization methods have been proposed to find a finite set of solutions to approximate the Pareto set and front (Miettinen, 1999; Ehrgott, 2005; Zhou et al., 2011). If at least $k$ solutions are needed to handle each dimension of the Pareto front, the required number of solutions could be $O(k^{(m-1)})$ for a problem with $m$ optimization objectives. Two illustrative examples with a set of solutions to approximate the Pareto front for problems with 2 and 3 objectives are shown in Figure 1. However, the required number of solutions will increase exponentially with the objective number $m$, leading to an extremely high computational overhead. It could also be very challenging for decision-makers to efficiently handle such a large set of solutions. Indeed, for a problem with many optimization objectives, a large portion of the solutions could become non-dominated and hence incomparable with each other (Purshouse & Fleming, 2007; Knowles & Corne, 2007).

In the past few decades, different heuristic and evolutionary algorithms have been proposed to tackle the many-objective black-box optimization problems (Zhang & Li, 2007; Bader & Zitzler, 2011; Deb & Jain, 2013). These algorithms typically aim to find a set of a few hundred solutions to handle problems with $4$ to a few dozen optimization objectives (Li et al., 2015; Sato & Ishibuchi, 2023). However, they still struggle to tackle problems with significantly many objectives (e.g., $> 100$), and cannot efficiently solve large-scale differentiable optimization problems. Dimensionality reduction (Deb & Saxena, 2005; Brockhoff & Zitzler, 2006; Singh et al., 2011) is a widely used technique to deal with many-objective optimization problems with potential redundant objectives. By summarizing all objectives by a few representative objectives, these methods can reformulate the originally challenging problem into a simpler problem with much fewer objectives. A detailed discussion with the dimensionality reduction can be found in Appendix D.6.

## 2.2 GRADIENT-BASED MULTI-OBJECTIVE OPTIMIZATION

When all objective functions are differentiable with gradient $\{\nabla f_i(\boldsymbol{x})\}_{i=1}^m$, we have the following definition for Pareto stationarity:

**Definition 4** (Pareto Stationary Solution). *A solution $\boldsymbol{x} \in \mathcal{X}$ is Pareto stationary if there exists a set of weights $\boldsymbol{\alpha} \in \boldsymbol{\Delta}^{m-1} = \{\boldsymbol{\alpha} | \sum_{i=1}^m \alpha_i = 1, \alpha_i \geq 0 \ \forall i\}$ such that the convex combination of gradients $\sum_{i=1}^m \alpha_i \nabla f_i(\boldsymbol{x}) = \boldsymbol{0}$.*

**Multiple Gradient Descent Algorithm**   One popular gradient-based approach is to find a valid gradient direction such that the values of all objective functions can be simultaneously improved (Fliege & Svaiter, 2000; Schäffler et al., 2002; Désidéri, 2012). The multiple gradient descent algorithm (MGDA) (Désidéri, 2012; Sener & Koltun, 2018) obtains a valid gradient $\boldsymbol{d}_t = \sum_{i=1}^m \alpha_i \nabla f_i(\boldsymbol{x})$ by solving the following quadratic programming problem at each iteration:

$$\min_{\alpha_i} || \sum_{i=1}^m \alpha_i \nabla f_i(\boldsymbol{x}_t)||_2^2, \quad s.t. \ \sum_{i=1}^m \alpha_i = 1, \quad \alpha_i \geq 0, \forall i = 1, ..., m, \tag{2}$$

and updates the current solution by a simple gradient descent $\boldsymbol{x}_{t+1} = \boldsymbol{x}_t - \eta_t \boldsymbol{d}_t$. If $\boldsymbol{d}_t = \boldsymbol{0}$, it means that there is no valid gradient direction that can improve all objectives at the same time, and therefore $\boldsymbol{x}_t$ is a Pareto stationary solution (Désidéri, 2012; Fliege et al., 2019). This idea has inspired many adaptive gradient methods for multi-task learning (Yu et al., 2020; Liu et al., 2021a;b; Momma et al., 2022; Liu et al., 2022; Navon et al., 2022; Senushkin et al., 2023; Lin et al., 2023; Liu et al., 2024). Different stochastic multiple gradient methods have also been proposed in recent years (Liu & Vicente, 2021; Zhou et al., 2022; Fernando et al., 2023; Chen et al., 2023; Xiao et al., 2023).

The location of solutions found by the original MGDA is not controllable, and several extensions have been proposed to find a set of diverse solutions with different trade-offs (Lin et al., 2019; Mahapatra & Rajan, 2020; Ma et al., 2020; Liu et al., 2021c). However, a large number of solutions is still required for a good approximation to the Pareto front. For solving problems with many objectives, MGDA and its extensions will suffer from a high computational overhead due to the high-dimensional quadratic programming problem (2) at each iteration. In addition, since a large portion of solutions is non-dominated with each other, it could be very hard to find a valid gradient direction to optimize all objectives at the same time.

**Scalarization Method**   Another popular class of methods for multi-objective optimization is the scalarization approach (Miettinen, 1999; Zhang & Li, 2007). The most straightforward method is

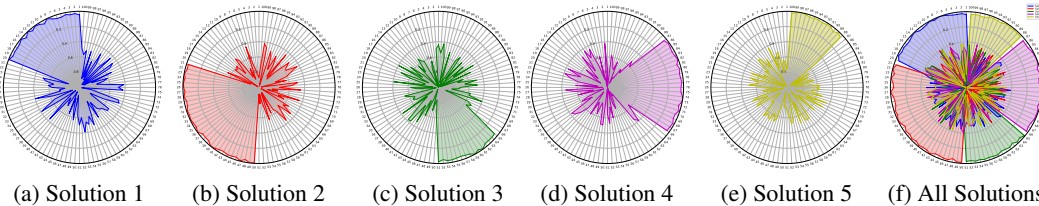

(a) Solution 1     (b) Solution 2     (c) Solution 3     (d) Solution 4     (e) Solution 5     (f) All Solutions

Figure 2: **Few Solutions to Address Many Optimization Objectives. (a)-(e):** 5 different solutions to tackle different optimization objectives in a complementary manner. **(f):** They together successfully handle all 100 optimization objectives.

linear scalarization (Geoffrion, 1967):

(Linear Scalarization)
$$\min_{\boldsymbol{x}\in\mathcal{X}} g^{(\text{LS})}(\boldsymbol{x}|\boldsymbol{\lambda}) = \sum_{i=1}^{m} \lambda_i f_i(\boldsymbol{x}), \tag{3}$$

where $\boldsymbol{\lambda} = (\lambda_1, \ldots, \lambda_m)$ is a preference vector over $m$ objectives on the simplex $\boldsymbol{\Delta}^{m-1} = \{\boldsymbol{\lambda} | \sum_{i=1}^{m} \lambda_i = 1, \lambda_i \geq 0 \ \forall i\}$. A set of diverse solutions can be obtained by solving the scalarization problem (3) with different preferences. Recently, different studies have shown that a well-tuned linear scalarization can outperform many adaptive gradient methods for multi-task learning Kurin et al. (2022); Xin et al. (2022); Lin et al. (2022); Royer et al. (2023). However, from the viewpoint of multi-objective optimization, linear scalarization cannot find any Pareto solution on the non-convex part of the Pareto front (Das & Dennis, 1997; Ehrgott, 2005; Hu et al., 2023).

Many other scalarization methods have been proposed in past decades. Among them, the Tchebycheff scalarization with good theoretical properties is a promising alternative (Bowman, 1976; Steuer & Choo, 1983):

(Tchebycheff Scalarization)
$$\min_{\boldsymbol{x}\in\mathcal{X}} g^{(\text{TCH})}(\boldsymbol{x}|\boldsymbol{\lambda}) = \max_{1 \leq i \leq m} \left\{ \lambda_i (f_i(\boldsymbol{x}) - z_i^*) \right\}, \tag{4}$$

where $\boldsymbol{\lambda} \in \boldsymbol{\Delta}^{m-1}$ is the preference and $\boldsymbol{z}^* \in \mathbb{R}^m$ is the ideal point (e.g., $z_i^* = \min f_i(\boldsymbol{x}) - \epsilon$ with a small $\epsilon > 0$). It is well-known that the Tchebycheff scalarization is able to find all weakly Pareto solutions for any Pareto front (Choo & Atkins, 1983). However, the $\max$ operator makes it become nonsmooth and hence suffers from a slow convergence rate by subgradient descent (Goffin, 1977) for differentiable multi-objective optimization. Recently, a smooth Tchebycheff scalarization approach (Lin et al., 2024) has been proposed to tackle the nonsmoothness issue:

(Smooth Tchebycheff Scalarization)
$$\min_{\boldsymbol{x}\in\mathcal{X}} g_\mu^{(\text{STCH})}(\boldsymbol{x}|\boldsymbol{\lambda}) = \mu \log \left( \sum_{i=1}^{m} e^{\frac{\lambda_i (f_i(\boldsymbol{x}) - z_i^*)}{\mu}} \right), \tag{5}$$

where $\mu$ is the smooth parameter with a small positive value (e.g., 0.1). According to (Lin et al., 2024), this smooth scalarization approach enjoys a fast convergence rate for the gradient-based method, while also having good theoretical properties for multi-objective optimization. A similar smooth optimization approach has also been proposed in He et al. (2024) for robust multi-task learning. Very recently, Qiu et al. (2024) prove and analyze the theoretical advantages of smooth Tchebycheff scalarization (5) over the classic Tchebycheff scalarization (4) for multi-objective reinforcement learning (MORL).

The scalarization methods do not have to solve a quadratic programming problem at each iteration and thus have lower pre-iteration complexity than MGDA. However, they still need to solve a large number of scalarization problems with different preferences to obtain a dense set of solutions to approximate the whole Pareto set.

## 3 TCHEBYCHEFF SET SCALARIZATION FOR MANY OBJECTIVE OPTIMIZATION

### 3.1 SMALL SOLUTION SET FOR MANY-OBJECTIVE OPTIMIZATION

Unlike previous methods, this work does not aim to find a huge set of solutions for approximating the whole Pareto set. Instead, we want to find a small set of solutions in a collaborative and

complementary way such that each optimization objective can be well addressed by at least one solution. We have the following formulation of our targeted set optimization problem:

$$\min_{\boldsymbol{X}_K=\{\boldsymbol{x}^{(k)}\}_{k=1}^K} \boldsymbol{f}(\boldsymbol{x}) = (\min_{\boldsymbol{x}\in\boldsymbol{X}_K} f_1(\boldsymbol{x}), \min_{\boldsymbol{x}\in\boldsymbol{X}_K} f_2(\boldsymbol{x}), \cdots, \min_{\boldsymbol{x}\in\boldsymbol{X}_K} f_m(\boldsymbol{x})), \qquad (6)$$

where $\boldsymbol{X}_K = \{\boldsymbol{x}^{(k)}\}_{k=1}^K$ is a set of $K$ solutions to tackle all $m$ objectives $\{f_i(\boldsymbol{x})\}_{i=1}^m$. With a large $K \geq m$, we will have a degenerated problem:

$$\min_{\boldsymbol{X}_K} \boldsymbol{f}(\boldsymbol{x}) = (\min_{\boldsymbol{x}^{(1)}\in\mathcal{X}} f_1(\boldsymbol{x}), \min_{\boldsymbol{x}^{(2)}\in\mathcal{X}} f_2(\boldsymbol{x}), \cdots, \min_{\boldsymbol{x}^{(m)}\in\mathcal{X}} f_m(\boldsymbol{x})), \qquad (7)$$

where each objective function $f_i$ is independently solved by its corresponding solution $\boldsymbol{x}^{(i)} \in \mathcal{X}$ via single objective optimization and the rest $(K - m)$ solutions are redundant. If $K = 1$, the set optimization problem (6) will be reduced to the standard multi-objective optimization problem (1).

In this work, we are more interested in the case $1 < K \ll m$, which finds a small set of solutions (e.g., $K = 5$) to tackle a large number of objectives (e.g., $m \geq 100$) as illustrated in Figure 2. In the ideal case, if the ground truth optimal objective group assignment is already known (e.g., which objectives should be optimized together by the same solution), it is straightforward to directly find an optimal solution for each group of objectives. However, for a general optimization problem, the ground truth objective group assignment is usually unknown in most cases, and finding the optimal assignment could be very difficult.

Very recently, a similar setting has been investigated in two concurrent works (Ding et al., 2024; Li et al., 2024). Ding et al. (2024) study the sum-of-minimum (SoM) optimization problem $\frac{1}{m}\sum_{i=1}^m \min\{f_i(\boldsymbol{x}^{(1)}), f_i(\boldsymbol{x}^{(2)}), \ldots, f_i(\boldsymbol{x}^{(K)})\}$ that can be found in many machine learning applications such as mixed linear regression (Yi et al., 2014; Zhong et al., 2016). They generalize the classic k-means++ (Arthur, 2007) and Lloyd's algorithm (Lloyd, 1982) for clustering to tackle this problem, but do not take multi-objective optimization into consideration. Li et al. (2024) propose a novel Many-objective multi-solution Transport (MosT) framework to tackle many-objective optimization. With a bi-level optimization formulation, they adaptively construct a few weighted multi-objective optimization problems that are assigned to different representative regions on the Pareto front. By solving these weighted problems with MGDA, a diverse set of solutions can be obtained to well cover all objectives. In this work, we propose a straightforward and efficient set scalarization approach to explicitly optimize all objectives by a small set of solutions. A detailed experimental comparison with these methods can be found in Section 4.

## 3.2 TCHEBYCHEFF SET SCALARIZATION

The set optimization formulation (6) is still a multi-objective optimization problem. In non-trivial cases, there is no single small solution set $\boldsymbol{X}_K$ with $K < m$ solutions that can optimize all $m$ objective functions $\{f_i(\boldsymbol{x})\}_{i=1}^m$ at the same time. To tackle this optimization problem, we propose the following Tchebycheff set (TCH-Set) scalarization approach:

$$\min_{\boldsymbol{X}_K=\{\boldsymbol{x}^{(k)}\}_{k=1}^K} g^{(\text{TCH-Set})}(\boldsymbol{X}_K|\boldsymbol{\lambda}) = \max_{1\leq i\leq m}\left\{\lambda_i(\min_{\boldsymbol{x}\in\boldsymbol{X}_K} f_i(\boldsymbol{x}) - z_i^*)\right\}$$

$$= \max_{1\leq i\leq m}\left\{\lambda_i(\min_{1\leq k\leq K} f_i(\boldsymbol{x}^{(k)}) - z_i^*)\right\}, \qquad (8)$$

where $\boldsymbol{\lambda} = (\lambda_1,\ldots,\lambda_m)$ and $\boldsymbol{z}^* = (z_1^*,\ldots,z_m^*)$ are the preference and ideal point for each objective function. In this way, all objective values $\{f_i(\boldsymbol{x})\}_{i=1}^m$ among the whole solution set $\boldsymbol{X}_K = \{\boldsymbol{x}^{(k)}\}_{k=1}^K$ are scalarized into a single function $g^{(\text{TCH-Set})}(\boldsymbol{X}_K|\boldsymbol{\lambda})$. In this work, a simple uniform vector $\boldsymbol{\lambda} = (\frac{1}{m},\ldots,\frac{1}{m})$ is used in all experiments without any specific preference among the objectives. A discussion on the effect of different preferences can be found in Appendix D.2.

By optimizing this TCH-Set scalarization function (8), we want to find an optimal small solution set $\boldsymbol{X}_K^*$ such that each objective can be well addressed by at least one solution $\boldsymbol{x}^{(k)} \in \boldsymbol{X}_K$ with a low worst objective value $\max_{1\leq i\leq m}\left\{\lambda_i(\min_{1\leq k\leq K} f_i(\boldsymbol{x}^{(k)}) - z_i^*)\right\}$. When the solution set contains only one single solution (e.g., $K = 1$), it will be reduced to the classic single-solution Tchebycheff scalarization (4). To avoid degenerated cases and focus on the key few-for-many setting, we make the following two assumptions in this work:

**Assumption 1** (No Redundant Solution). *We assume that no solution in the optimal solution set* $X_K^* = \texttt{argmin}_{X_k} g^{(TCH\text{-}Set)}(X_K|\lambda)$ *will be redundant, which means*

$$g^{(TCH\text{-}Set)}(X_K^*|\lambda) < g^{(TCH\text{-}Set)}(X_K^* \setminus \{x^{(k)}\}|\lambda) \tag{9}$$

*holds for any* $1 \le k \le K \le m$ *with* $\lambda > \mathbf{0}$.

**Assumption 2** (All Positive Preference). *In the few-for-many setting, we assume that the preferences should be positive for all objectives such that* $\{\sum_{i=1}^m \lambda_i = 1, \lambda_i > 0 \; \forall i\}$.

These assumptions are reasonable in practice, especially for the few-for-many setting (e.g., $K \ll m$) considered in this work. For a general problem, it is extremely rare that a small number of solutions (e.g., $4$) can exactly solve a large number of objectives (e.g., $m = 1,000$). From the viewpoint of clustering (Ding et al., 2024), it is analogous to the case that all the $1,000$ data points under consideration are exactly located in $4$ different locations. If we do meet this extreme case with $p$ redundant solutions, we can simply reduce the number of solutions to make it a $(K - p)$-solutions-for-$m$-objectives problem. Similarly, if some of the preferences (e.g., $q$) are $0$, we can also simply reduce the original $m$-objective problem into a $(m - q)$-objective problem that satisfies the positive preference assumption.

From the viewpoint of multi-objective optimization, we have the following guarantee for the optimal solution set of Tchebycheff set scalarization:

**Theorem 1** (Existence of Pareto Optimal Solution for Tchebycheff Set Scalarization). *There exists an optimal solution set* $\bar{X}_K^*$ *for the Tchebycheff set scalarization optimization problem (8) such that all solutions in* $\bar{X}_K^*$ *are Pareto optimal of the original multi-objective optimization problem (1). In addition, if the optimal set* $X_K^*$ *is unique, all solutions in* $X_K^*$ *are Pareto optimal.*

*Proof Sketch.* This theorem can be proved by construction and contradiction based on Definition 2 for (weakly) Pareto optimality and the form of Tchebycheff set Scalarization (8). A detailed proof is provided in Appendix A.1. □

It should be emphasized that, without the strong unique optimal solution set assumption, we only have a weak existence guarantee for the Pareto optimality. For an optimal solution set for the Tchebycheff set scalarization (8), it is possible that many of the solutions are not even weakly Pareto optimal for the original multi-objective optimization problem (1). This finding could be a bit surprising for multi-objective optimization, and we provide a detailed discussion in Appendix A.5. In addition to the Pareto optimality guarantee, the non-smoothness of TCH-Set scalarization might also hinder its practical usage for efficient optimization. To address these crucial issues, we further develop a smooth Tchebycheff set scalarization with good optimization property and promising Pareto optimality guarantee in the next subsection.

### 3.3 SMOOTH TCHEBYCHEFF SET SCALARIZATION

The Tchebycheff set scalarization formulation (8) involves a $\max$ and a $\min$ operator, which leads to its non-smoothness even when all objective functions $\{f_i(\boldsymbol{x})\}_{i=1}^m$ are smooth. In other words, the Tchebycheff set scalarization $g^{(TCH\text{-}Set)}(\boldsymbol{X}_K|\boldsymbol{\lambda})$ is not differentiable and will suffer from a slow convergence rate by subgradient descent. To address this issue, we leverage the smooth optimization approach (Nesterov, 2005; Beck & Teboulle, 2012; Chen, 2012) to propose a smooth Tchebycheff set scalarization for multi-objective optimization.

According to Beck & Teboulle (2012), for the maximization function among all objectives $\max_{1 \le i \le m}\{f_1(\boldsymbol{x}), f_2(\boldsymbol{x}), \ldots, f_m(\boldsymbol{x})\}$, we have the smooth maximization function:

$$\operatorname*{smax}_{1 \le i \le m}\{f_1(\boldsymbol{x}), f_2(\boldsymbol{x}), \ldots, f_m(\boldsymbol{x})\} = \mu \log \left(\sum_{i=1}^m e^{\frac{f_i(\boldsymbol{x})}{\mu}}\right), \tag{10}$$

where $\mu$ is a smooth parameter. Similarly, for the minimization among different solutions for the $i$-th objective $\min_{1 \le k \le K}\{f_i(\boldsymbol{x}^{(1)}), f_i(\boldsymbol{x}^{(2)}), \ldots, f_i(\boldsymbol{x}^{(K)})\}$, we have the smooth minimization function:

$$\operatorname*{smin}_{1 \le k \le K}\{f_i(\boldsymbol{x}^{(1)}), f_i(\boldsymbol{x}^{(2)}), \ldots, f_i(\boldsymbol{x}^{(K)})\} = -\mu_i \log \left(\sum_{k=1}^K e^{-\frac{f_i(\boldsymbol{x}^{(k)})}{\mu_i}}\right), \tag{11}$$

where $\mu_i$ is a smooth parameter.

By leveraging the above smooth maximization and minimization functions, we propose the smooth Tchebycheff set scalarization (STCH-Set) scalarization for multi-objective optimization:

$$\min_{\boldsymbol{X}_K = \{\boldsymbol{x}^{(k)}\}_{k=1}^K} g_{\mu,\{\mu_i\}_{i=1}^m}^{(\text{STCH-Set})}(\boldsymbol{X}_K | \boldsymbol{\lambda}) = \operatorname*{smax}_{1 \le i \le m} \left\{ \lambda_i \operatorname*{smin}_{1 \le k \le K} f_i(\boldsymbol{x}^{(k)}) - z_i^* \right\}$$

$$= \mu \log \left( \sum_{i=1}^m e^{\frac{\lambda_i \left( \operatorname*{smin}_{1 \le k \le K} f_i(\boldsymbol{x}^{(k)}) - z_i^* \right)}{\mu}} \right)$$

$$= \mu \log \left( \sum_{i=1}^m e^{\frac{\lambda_i \left( -\mu_i \log \left( \sum_{k=1}^K e^{-f_i(\boldsymbol{x}^{(k)})/\mu_i} \right) - z_i^* \right)}{\mu}} \right). \qquad (12)$$

If the same smooth parameter $\mu_1 = \ldots = \mu_m = \mu$ are used for all smooth terms, we have the following simplified formulation:

$$\min_{\boldsymbol{X}_K = \{\boldsymbol{x}^{(k)}\}_{k=1}^K} g_{\mu}^{(\text{STCH-Set})}(\boldsymbol{X}_K | \boldsymbol{\lambda}) = \mu \log \left( \sum_{i=1}^m e^{\lambda_i \left( -\log \left( \sum_{k=1}^K e^{-f_i(\boldsymbol{x}^{(k)})/\mu} \right) - z_i^* \right)} \right). \qquad (13)$$

When $K = 1$, it will reduce to the single-solution smooth Tchebycheff scalarization (5). Similarly to the non-smooth TCH-Set counterpart, the STCH-Set scalarization also has good theoretical properties for multi-objective optimization:

**Theorem 2** (Pareto Optimality for Smooth Tchebycheff Set Scalarization). *All solutions in the optimal solution set $\boldsymbol{X}_K^*$ for the smooth Tchebycheff set scalarization problem (12) are weakly Pareto optimal of the original multi-objective optimization problem (1). In addition, the solutions are Pareto optimal if either*

1. *the optimal solution set $\boldsymbol{X}_K^*$ is unique, or*

2. *all preference coefficients are positive ($\lambda_i > 0 \ \forall i$).*

*Proof Sketch.* This theorem can be proved by contradiction based on Definition 2 for (weakly) Pareto optimality and the form of smooth Tchebycheff set scalarization (12). We provide a detailed proof in Appendix A.2. □

**Theorem 3** (Uniform Smooth Approximation). *The smooth Tchebycheff set (STCH-Set) scalarization $g_{\mu,\{\mu_i\}_{i=1}^m}^{(\text{STCH-Set})}(\boldsymbol{X}_K | \boldsymbol{\lambda})$ is a uniform smooth approximation of the Tchebycheff set (TCH-Set) scalarization $g^{(\text{TCH-Set})}(\boldsymbol{X}_K | \boldsymbol{\lambda})$, and we have:*

$$\lim_{\mu \downarrow 0, \mu_i \downarrow 0 \ \forall i} g_{\mu,\{\mu_i\}_{i=1}^m}^{(\text{STCH-Set})}(\boldsymbol{X}_K | \boldsymbol{\lambda}) = g^{(\text{TCH-Set})}(\boldsymbol{X}_K | \boldsymbol{\lambda}) \qquad (14)$$

*for any valid set $\boldsymbol{X}_k \subset \mathcal{X}$.*

*Proof Sketch.* This theorem can be proved by deriving the upper and lower bounds of the TCH-Set scalarization $g^{(\text{TCH-Set})}(\boldsymbol{X}_K | \boldsymbol{\lambda})$ with respect to the smooth smax and smin operators in the STCH-Set scalarization $g_{\mu,\{\mu_i\}_{i=1}^m}^{(\text{STCH-Set})}(\boldsymbol{X}_K | \boldsymbol{\lambda})$. A detailed proof is provided in Appendix A.3. □

According to these theorems, for any smooth parameters $\mu$ and $\{\mu\}_{i=1}^m$, all solutions in the optimal set $\boldsymbol{X}_K^*$ for STCH-Set scalarization (12) are also (weakly) Pareto optimal. In addition, with small smooth parameters $\mu \downarrow 0, \mu_i \downarrow 0 \ \forall i$, the value of $g_{\mu,\{\mu_i\}_{i=1}^m}^{(\text{STCH-Set})}(\boldsymbol{X}_K | \boldsymbol{\lambda})$ will be close to its non-smooth counterpart $g^{(\text{TCH-Set})}(\boldsymbol{X}_K | \boldsymbol{\lambda})$ for all valid solution sets $\boldsymbol{X}_K$, which is also the case for the optimal set $\boldsymbol{X}_K^* = \arg\min_{\boldsymbol{X}_K} g^{(\text{TCH-Set})}(\boldsymbol{X}_K | \boldsymbol{\lambda})$. Therefore, it is reasonable to find an approximate optimal solution set for the original $g^{(\text{TCH-Set})}(\boldsymbol{X}_K | \boldsymbol{\lambda})$ by optimizing its smooth counterpart $g_{\mu,\{\mu_i\}_{i=1}^m}^{(\text{STCH-Set})}(\boldsymbol{X}_K | \boldsymbol{\lambda})$ with small smooth parameters $\mu$ and $\{\mu_i\}_{i=1}^m$.

---

**Algorithm 1** STCH-Set Scalarization for Multi-Objective Optimization

---

1: **Input:** Preference $\boldsymbol{\lambda}$, Step Size $\{\eta_t\}_{t=0}^T$, Initial $\boldsymbol{X}_K^0$
2: **for** $t = 1$ to $T$ **do**
3:     $\boldsymbol{X}_K^t = \boldsymbol{X}_K^{t-1} - \eta_t \nabla g^{\text{(STCH-Set)}}(\boldsymbol{X}_K^{t-1}|\boldsymbol{\lambda})$
4: **end for**
5: **Output:** Final Solution $\boldsymbol{X}_K^T$

---

The STCH-Set scalarization function $g_{\mu,\{\mu_i\}_{i=1}^m}^{\text{(STCH-Set)}}(\boldsymbol{X}_K|\boldsymbol{\lambda})$ can be efficiently optimized by any gradient-based optimization method, where we treat all solutions as a single solution matrix. A simple gradient descent algorithm for STCH-Set is shown in **Algorithm 1**. However, when the objective functions $\{f_i(\boldsymbol{x})\}_{i=1}^m$ are highly non-convex, it could be very hard to find the global optimal set $\boldsymbol{X}_K^*$ for $g_{\mu,\{\mu_i\}_{i=1}^m}^{\text{(STCH-Set)}}(\boldsymbol{X}_K|\boldsymbol{\lambda})$. In this case, we have the Pareto stationarity guarantee for the gradient-based method:

**Theorem 4** (Convergence to Pareto Stationary Solution). *If there exists a solution set $\hat{\boldsymbol{X}}_K$ such that $\nabla_{\hat{\boldsymbol{x}}^{(k)}} g_{\mu,\{\mu_i\}_{i=1}^m}^{\text{(STCH-Set)}}(\hat{\boldsymbol{X}}_K|\boldsymbol{\lambda}) = \boldsymbol{0}$ for all $\hat{\boldsymbol{x}}^{(k)} \in \hat{\boldsymbol{X}}_K$, then all solutions in $\hat{\boldsymbol{X}}_K$ are Pareto stationary solutions of the original multi-objective optimization problem (1).*

*Proof Sketch.* We can prove this theorem by analyzing the form of gradient for STCH-Set scalarization $\nabla_{\hat{\boldsymbol{x}}^{(k)}} g_{\mu,\{\mu_i\}_{i=1}^m}^{\text{(STCH-Set)}}(\hat{\boldsymbol{X}}_K|\boldsymbol{\lambda}) = \boldsymbol{0}$ for each solution $\boldsymbol{x}^{(k)}$ with the condition for Pareto stationarity in Definition 4. A detailed proof is provided in Appendix A.4. □

When $K$ reduces to 1, all these theorems exactly match their counterpart theorems for single-solution smooth Tchebycheff scalarization (Lin et al., 2024).

## 4 EXPERIMENTS

**Baseline Methods** In this section, we compare our proposed TCH-Set and STCH-Set scalarization with three simple scalarization methods: (1) linear scalarization (LS) with randomly sampled preferences, (2) Tchebycheff scalarization (TCH) with randomly sampled preferences, (3) smooth Tchebycheff scalarization (STCH) with randomly sampled preferences (Lin et al., 2024), as well as two recently proposed methods for finding a small set of solutions: (4) the many-objective multi-solution transport (MosT) method (Li et al., 2024), and (5) the efficient sum-of-minimum (SoM) optimization method (Ding et al., 2024).

**Experimental Setting** This work aims to find a small set $\boldsymbol{X}_K$ with $K$ solutions to address all $m$ objectives in each problem. For each objective $f_i(\boldsymbol{x})$, we care about the best objective value $\min_{\boldsymbol{x} \in \boldsymbol{X}_K} f_i(\boldsymbol{x})$ achieved by the solutions in $\boldsymbol{X}_K$. Both the worst obtained value among all objectives $\max_{1 \le i \le m} \min_{\boldsymbol{x} \in \boldsymbol{X}_K} f_i(\boldsymbol{x})$ and the average objective value $\frac{1}{m} \sum_{i=1}^m \min_{\boldsymbol{x} \in \boldsymbol{X}_K} f_i(\boldsymbol{x})$ are reported for comparison. Detailed experimental settings for each problem can be found in Appendix B. Due to the page limit, more experimental results and analysis can be found in Appendix C.

### 4.1 EXPERIMENTAL RESULTS AND ANALYSIS

**Convex Many-Objective Optimization** We first test our proposed methods for solving convex multi-objective optimization with $m = 128$ or $m = 1,024$ objective functions with different numbers of solutions (from 3 to 20). We independently run each comparison 50 times. In each run, we randomly generate $m$ independent convex quadratic functions as the optimization objectives for all methods. The mean worst and average objective values for each method over 50 runs on the $m = 128$ case are reported in Table 1 and the full table can be found in Appendix C.1. We also conduct the Wilcoxon rank-sum test between STCH-Set and other methods for all comparisons at the 0.05 significant level. The $(+/=/-)$ symbol means the result obtained by STCH-Set is significantly better, equal to, or worse than the results of the compared method.

Table 1: The results on convex optimization problems with $K = \{3, 4, 5, 6, 8, 10\}$ solutions. Mean worst and average objective values over 50 runs are reported. Best results are highlighted in **bold**, and the full table can be found in Appendix C.1.

| | | LS | TCH | STCH | MosT | SoM | TCH-Set | STCH-Set |
|---|---|---|---|---|---|---|---|---|
| | | | | number of objectives $m = 128$ | | | | |
| $K = 3$ | worst | 4.64e+00(+) | 4.44e+00(+) | 4.41e+00(+) | 2.12e+00(+) | 1.86e+00(+) | 1.02e+00(+) | **6.08e-01** |
| | average | 8.61e-01(+) | 8.03e-01(+) | 7.80e-01(+) | 3.45e-01(+) | **2.02e-01(-)** | 3.46e-01(+) | 2.12e-01 |
| $K = 4$ | worst | 4.23e+00(+) | 3.83e+00(+) | 3.68e+00(+) | 1.48e+00(+) | 1.12e+00(+) | 6.74e-01(+) | **3.13e-01** |
| | average | 7.45e-01(+) | 6.54e-01(+) | 6.41e-01(+) | 1.85e-01(+) | 1.12e-01(+) | 2.27e-01(+) | **9.44e-02** |
| $K = 5$ | worst | 4.17e+00(+) | 3.56e+00(+) | 3.51e+00(+) | 1.02e+00(+) | 9.20e-01(+) | 4.94e-01(+) | **1.91e-01** |
| | average | 7.08e-01(+) | 5.75e-01(+) | 5.05e-01(+) | 9.81e-02(+) | 7.95e-02(+) | 1.60e-01(+) | **5.12e-02** |
| $K = 6$ | worst | 3.81e+00(+) | 3.41e+00(+) | 3.20e+00(+) | 9.56e-01(+) | 7.03e-01(+) | 3.72e-01(+) | **1.36e-01** |
| | average | 6.72e-01(+) | 5.31e-01(+) | 5.15e-01(+) | 8.34e-02(+) | 5.34e-02(+) | 1.19e-01(+) | **3.15e-02** |
| $K = 8$ | worst | 3.69e+00(+) | 2.94e+00(+) | 2.61e+00(+) | 8.32e-01(+) | 5.20e-01(+) | 2.84e-01(+) | **1.02e-01** |
| | average | 6.16e-01(+) | 4.57e-01(+) | 4.22e-01(+) | 6.91e-02(+) | 3.40e-02(+) | 8.53e-02(+) | **1.78e-02** |
| $K = 10$ | worst | 3.68e+00(+) | 2.69e+00(+) | 2.40e+00(+) | 6.27e-01(+) | 4.07e-01(+) | 1.95e-01(+) | **7.99e-02** |
| | average | 5.69e-01(+) | 4.07e-01(+) | 3.20e-01(+) | 4.59e-02(+) | 2.43e-02(+) | 5.92e-02(+) | **1.34e-02** |
| | | | | Wilcoxon Rank-Sum Test Summary | | | | |
| $+/ = /-$ | worst | 6/0/0 | 6/0/0 | 6/0/0 | 6/0/0 | 6/0/0 | 6/0/0 | - |
| | average | 6/0/0 | 6/0/0 | 6/0/0 | 6/0/0 | 5/0/1 | 6/0/0 | - |

Table 2: The results on mixed linear regression with noisy level $\sigma = 0.1$. Full table in Appendix C.2.

| | | LS | TCH | STCH | SoM | TCH-Set | STCH-Set |
|---|---|---|---|---|---|---|---|
| $K = 5$ | worst | 3.84e+01(+) | 2.45e+01(+) | 2.43e+01(+) | 2.50e+00(+) | 4.19e+00(+) | **2.10e+00** |
| | average | 2.70e+00(+) | 1.46e+00(+) | 1.44e+00(+) | **1.88e-01(-)** | 6.53e-01(+) | 4.72e-01 |
| $K = 10$ | worst | 2.21e+01(+) | 2.49e+01(+) | 3.92e+01(+) | 3.46e+00(+) | 2.04e+00(+) | **5.00e-01** |
| | average | 1.09e+00(+) | 1.19e+00(+) | 3.01e+00(+) | 2.33e-01(+) | 3.83e-01(+) | **2.00e-01** |
| $K = 15$ | worst | 4.20e+01(+) | 2.11e+01(+) | 2.21e+01(+) | 2.57e+00(+) | 1.64e+00(+) | **2.70e-01** |
| | average | 3.07e+00(+) | 1.02e+00(+) | 1.02e+00(+) | 1.97e-01(+) | 3.22e-01(+) | **1.66e-01** |
| $K = 20$ | worst | 4.20e+01(+) | 2.14e+01(+) | 2.04e+01(+) | 1.44e+00(+) | 1.79e+00(+) | **2.27e-01** |
| | average | 3.09e+00(+) | 8.35e-01(+) | 8.87e-01(+) | 1.84e-01(+) | 3.29e-01(+) | **1.66e-01** |

According to the results, the traditional methods (e.g., LS/TCH/STCH) fail to properly tackle the few solutions for many objectives setting considered in this work. MosT achieves much better performance than the traditional methods by actively finding a set of diverse solutions to cover all objectives, but it cannot tackle the problems with 1024 objectives in a reasonable time. In addition, MosT is outperformed by SoM and our proposed STCH-Set, which directly optimizes the performance for each objective. Our proposed STCH-Set performs the best in achieving the low worst objective values for all comparisons. In addition, although not explicitly designed, STCH-Set also achieves a very promising mean average performance for most comparisons since all objectives are well addressed. The importance of smoothness for set optimization is fully confirmed by the observation that STCH-Set significantly outperforms TCH-Set on all comparisons. More discussion on why STCH-Set can outperform SoM on the average performance can be found in Appendix D.4.

**Noisy Mixed Linear Regression** We then test different methods' performance on the noisy mixed linear regression problem as in (Ding et al., 2024). For each comparison, $1,000$ data points are randomly generated from $K$ ground truth linear models with noise. Then, with different optimization methods, we train $K$ linear regression models to tackle all $1,000$ data points where the objectives are the squared error for each point. Each comparison are run 50 times, and the detailed experiment setting is in Appendix B.

We conduct comparison with different numbers of $K = \{5, 10, 15, 20\}$ and noise levels $\sigma = \{0.1, 0.5, 1.0\}$. The results with $\sigma = 0.1$ are shown in Table 2, and the full results can be found in Appendix C.2. According to the results, our proposed STCH-Set can always achieve the lowest worst objective value, and achieves the best average objective value in most comparisons.

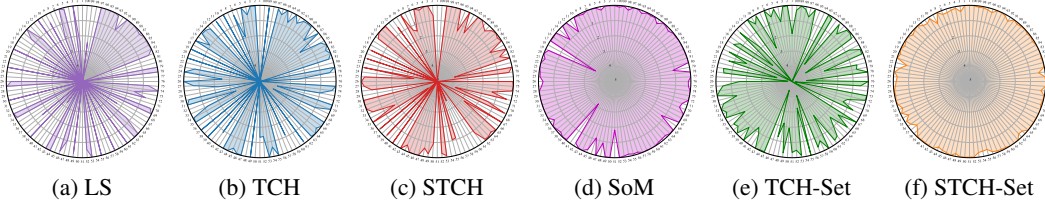

|  (a) LS | (b) TCH | (c) STCH | (d) SoM | (e) TCH-Set | (f) STCH-Set |

Figure 3: Different methods' performance for the same mixed nonlinear regression problem. We report the performance on the same set of 100 randomly sampled objectives. STCH-Set can properly address all objectives and achieve the best overall performance. TCH-Set has a much better worst objective value than LS/TCH/STCH but is not reflected in this figure.

Table 3: Results on mixed nonlinear regression with noisy level $\sigma = 0.1$. Full table in Appendix C.3.

|  |  | LS | TCH | STCH | SoM | TCH-Set | STCH-Set |
|---|---|---|---|---|---|---|---|
| $K = 5$ | worst | 2.34e+02 (+) | 1.68e+02(+) | 1.57e+02(+) | 1.68e+01(+) | 1.94e+01(+) | **7.43e+00** |
|  | average | 9.19e+00(+) | 8.50e+00(+) | 7.90e+00(+) | **8.54e-01(-)** | 3.48e+00(+) | 1.89e+00 |
| $K = 10$ | worst | 3.23e+02(+) | 1.72e+02(+) | 1.57e+02(+) | 4.42e+00(+) | 6.07e+00(+) | **6.28e-01** |
|  | average | 8.70e+00 (+) | 6.65e+00(+) | 5.85e+00(+) | 1.22e-01(+) | 1.03e+00(+) | **5.99e-02** |
| $K = 15$ | worst | 3.65e+02(+) | 2.10e+02(+) | 1.62e+02(+) | 1.10e+00(+) | 1.57e+00(+) | **2.05e-01** |
|  | average | 8.33e+00(+) | 5.66e+00(+) | 4.89e+00(+) | 1.47e-01(+) | 3.27e-01(+) | **1.27e-02** |
| $K = 20$ | worst | 3.36e+02(+) | 2.04e+02(+) | 1.81e+02(+) | 5.59e+00(+) | 4.37e+00(+) | **6.22e-01** |
|  | average | 8.81e+00(+) | 6.92e+00(+) | 6.23e+00(+) | 1.24e-01(+) | 9.09e-01(+) | **6.27e-02** |

**Noisy Mixed Nonlinear Regression** Following (Ding et al., 2024), we also compare different methods on the noisy mixed nonlinear regression. The problem setting (details in Appendix B) is similar to the mixed linear regression, but we now build nonlinear neural networks as the models. According to the results in Table 3 and Appendix C.3, STCH-Set can still obtain the best overall performance for most comparisons. We visualize different methods' performance on 100 sampled data in Figure 3, and it is clear that STCH-Set can well address all objectives with the best overall performance.

## 5 CONCLUSION, LIMITATION AND FUTURE WORK

**Conclusion** In this work, we have proposed a novel Tchebycheff set (TCH-Set) scalarization and a smooth Tchebycheff set (STCH-Set) scalarization to find a small set of solutions for many-objective optimization. When properly optimized, each objective in the original problem should be well addressed by at least one solution in the found solution set. Both theoretical analysis and experimental studies have been conducted to demonstrate the promising properties and efficiency of TCH-Set/STCH-Set scalarization. Our proposed method is a complement rather than a replacement for the traditional multi-objective optimization method, and we hope it can inspire more interesting follow-up work to tackle the important few-for-many problems that naturally arise in many real-world applications.

**Limitation and Future Work** This work proposes a general optimization method for multi-objective optimization which are not tied to particular applications. We do not see any specific potential societal impact of the proposed methods. This work only focuses on the deterministic optimization setting that all objectives are always available. One potential future research direction is to investigate how to deal with only partially observable objective values in practice.

ACKNOWLEDGMENTS

This work was supported by the Research Grants Council of the Hong Kong Special Administrative Region, China (CityU11215622 and CityU11215723), the National Natural Science Foundation of China (Grant No. 62476118), the Natural Science Foundation of Guangdong Province (Grant No. 2024A1515011759), and the National Natural Science Foundation of Shenzhen (Grant No.JCYJ20220530113013031).

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

In this appendix, we mainly provide:

- **Detailed Proofs and Discussion** for the theoretical analysis are provide in Section A.
- **Problem and Experimental Settings** can be found in Section B.
- **More Experimental Results and Analyses** are provided in Section C.
- **Ablation Studies and Discussions** can be found in Section D.

## A    DETAILED PROOF AND DISCUSSION

### A.1    PROOF OF THEOREM 1

**Theorem 1** (Existence of Pareto Optimal Solution for Tchebycheff Set Scalarization). *There exists an optimal solution set $\bar{\boldsymbol{X}}_K^*$ for the Tchebycheff set scalarization optimization problem (8) such that all solutions in $\bar{\boldsymbol{X}}_K^*$ are Pareto optimal of the original multi-objective optimization problem (1). In addition, if the optimal set $\boldsymbol{X}_K^*$ is unique, all solutions in $\boldsymbol{X}_K^*$ are Pareto optimal.*

*Proof.* This theorem can be proved by construction and contradiction based on Definition 2 for (weakly) Pareto optimality and the form of Tchebycheff set Scalarization (8).

**Existence of Pareto Optimal Solution**    We first prove that there exists an optimal solution set of which all solutions are Pareto optimal by construction.

Step 1: Let $\bar{\boldsymbol{X}}_K$ be an optimal solution set for the Tchebycheff set scalarization problem, we have:

$$\bar{\boldsymbol{X}}_K = \mathtt{argmin}_{\boldsymbol{X}_K} \max_{1 \le i \le m} \{\lambda_i (\min_{\boldsymbol{x} \in \boldsymbol{X}_K} f_i(\boldsymbol{x}) - \boldsymbol{z}_i^*)\}, \tag{15}$$

where $\bar{\boldsymbol{X}}_K = \{\bar{\boldsymbol{x}}^{(k)}\}_{k=1}^K$.

Step 2: Without loss of generality, we suppose the $k$-th solution $\bar{\boldsymbol{x}}^{(k)}$ in $\bar{\boldsymbol{X}}_K$ is not Pareto optimal, which means there exists a valid solution $\hat{\boldsymbol{x}} \in \mathcal{X}$ such that $f(\hat{\boldsymbol{x}}) \prec f(\bar{\boldsymbol{x}}^{(k)})$. In other words, we have:

$$f_i(\hat{\boldsymbol{x}}) \le f_i(\bar{\boldsymbol{x}}^{(k)}), \forall i \in \{1, ..., m\} \text{ and } f_j(\hat{\boldsymbol{x}}) < f_j(\bar{\boldsymbol{x}}^{(k)}), \exists j \in \{1, ..., m\}. \tag{16}$$

Let $\hat{\boldsymbol{X}}_K = (\bar{\boldsymbol{X}}_K \setminus \{\bar{\boldsymbol{x}}^{(k)}\}) \cup \{\hat{\boldsymbol{x}}\}$, we have:

$$\max_{1 \le i \le m} \{\lambda_i (\min_{\boldsymbol{x} \in \hat{\boldsymbol{X}}_K} f_i(\boldsymbol{x}) - \boldsymbol{z}_i^*)\} \le \max_{1 \le i \le m} \{\lambda_i (\min_{\boldsymbol{x} \in \bar{\boldsymbol{X}}_K} f_i(\boldsymbol{x}) - \boldsymbol{z}_i^*)\}. \tag{17}$$

Since $\bar{\boldsymbol{X}}_K$ is already the optimal solution set for Tchebycheff set scalarization, we have:

$$\max_{1 \le i \le m} \{\boldsymbol{\lambda}_i (\min_{\boldsymbol{x} \in \hat{\boldsymbol{X}}_K} f_i(\boldsymbol{x}) - \boldsymbol{z}_i^*)\} = \max_{1 \le i \le m} \{\lambda_i (\min_{\boldsymbol{x} \in \bar{\boldsymbol{X}}_K} f_i(\boldsymbol{x}) - \boldsymbol{z}_i^*)\}. \tag{18}$$

We treat $\hat{\boldsymbol{X}}_K$ as the current optimal solution set.

Step 3: Repeat Step 2 until no new solution $\hat{\boldsymbol{x}} \in \mathcal{X}$ can be found to dominate and replace any solution in the current solution set, and let $\hat{\boldsymbol{X}}_K^*$ be the final found optimal solution set for TCH-Set scalarization.

According to the definition of Pareto optimality, all solutions in the optimal solution set $\hat{\boldsymbol{X}}_K^*$ should be Pareto optimal.

**Pareto Optimality for Unique Optimal Solution Set**    This guarantee can be proved by contradiction based on definition for Pareto optimality, the form of TCH-Set Scalarization, and the uniqueness of the optimal solution set.

Let $\boldsymbol{X}_K^*$ be an optimal solution set for the Tchebycheff set scalarization problem, we have:

$$\boldsymbol{X}_K^* = \mathtt{argmin}_{\boldsymbol{X}_K} \max_{1 \le i \le m} \{\lambda_i (\min_{\boldsymbol{x} \in \boldsymbol{X}_K} f_i(\boldsymbol{x}) - \boldsymbol{z}_i^*)\}, \tag{19}$$

where $\boldsymbol{X}_K^* = \{\boldsymbol{x}^{*(k)}\}_{k=1}^K$.

Without loss of generality, we suppose the $k$-th solution $\boldsymbol{x}^{*(k)}$ in $\boldsymbol{X}_K^*$ is not Pareto optimal, which means there exists a valid solution $\hat{\boldsymbol{x}} \in \mathcal{X}$ such that $f(\hat{\boldsymbol{x}}) \prec f(\boldsymbol{x}^{*(k)})$. In other words, we have:

$$f_i(\hat{\boldsymbol{x}}) \leq f_i(\boldsymbol{x}^{*(k)}), \forall i \in \{1, ..., m\} \text{ and } f_j(\hat{\boldsymbol{x}}) < f_j(\boldsymbol{x}^{*(k)}), \exists j \in \{1, ..., m\}. \tag{20}$$

Let $\hat{\boldsymbol{X}}_K = (\boldsymbol{X}_K^* \setminus \{\boldsymbol{x}^{*(k)}\}) \cup \{\hat{\boldsymbol{x}}\}$, we have:

$$\max_{1 \leq i \leq m} \{\lambda_i(\min_{\boldsymbol{x} \in \hat{\boldsymbol{X}}_K} f_i(\boldsymbol{x}) - z_i^*)\} \leq \max_{1 \leq i \leq m} \{\lambda_i(\min_{\boldsymbol{x} \in \boldsymbol{X}_K^*} f_i(\boldsymbol{x}) - z_i^*)\}. \tag{21}$$

On the other hand, according to the uniqueness of $\boldsymbol{X}_K^*$, we have:

$$\max_{1 \leq i \leq m} \{\lambda_i(\min_{\boldsymbol{x} \in \hat{\boldsymbol{X}}_K} f_i(\boldsymbol{x}) - z_i^*)\} > \max_{1 \leq i \leq m} \{\lambda_i(\min_{\boldsymbol{x} \in \boldsymbol{X}_K^*} f_i(\boldsymbol{x}) - z_i^*)\}. \tag{22}$$

The above two inequalities contradict each other. Therefore, every solution $\boldsymbol{x}^{*(k)}$ in the unique optimal solution set $\boldsymbol{X}_K^*$ should be Pareto optimal.

$$\square$$

### A.2 PROOF OF THEOREM 2

**Theorem 2** (Pareto Optimality for Smooth Tchebycheff Set Scalarization). *All solutions in the optimal solution set $\boldsymbol{X}_K^*$ for the smooth Tchebycheff set scalarization problem (12) are weakly Pareto optimal of the original multi-objective optimization problem (1). In addition, the solutions are Pareto optimal if either*

1. *the optimal solution set $\boldsymbol{X}_K^*$ is unique, or*
2. *all preference coefficients are positive ($\lambda_i > 0 \; \forall i$).*

*Proof.* Similarly to the above proof for Theorem 1, this theorem can be proved by contradiction based on Definition 2 for (weakly) Pareto optimality and the form of smooth Tchebycheff set scalarization (12).

**Weakly Pareto Optimality** We first prove all the solutions in $\boldsymbol{X}_K^*$ are weakly Pareto optimal. Let $\boldsymbol{X}_K^*$ be an optimal solution set for the smooth Tchebycheff set scalarization problem (12), we have:

$$\boldsymbol{X}_K^* = \arg\min_{\boldsymbol{X}_K} \mu \log \left( \sum_{i=1}^m e^{\frac{-\mu_i \log\left(\sum_{k=1}^K e^{-\lambda_i(f_i(\boldsymbol{x}^{(k)}) - z_i^*)/\mu_i}\right)}{\mu}} \right), \tag{23}$$

where $\boldsymbol{X}_K^* = \{\boldsymbol{x}^{*(k)}\}_{k=1}^K$.

Without loss of generality, we further suppose the $k$-th solution $\boldsymbol{x}^{*(k)}$ in $\boldsymbol{X}_K^*$ is not weakly Pareto optimal for the original multi-objective optimization problem (1). According to Definition 2 for weakly Pareto optimality, there exists a valid solution $\hat{\boldsymbol{x}} \in \mathcal{X}$ such that $\boldsymbol{f}(\hat{\boldsymbol{x}}) \prec_{\text{strict}} \boldsymbol{f}(\boldsymbol{x}^{*(k)})$. In other words, we have:

$$f_i(\hat{\boldsymbol{x}}) < f_i(\boldsymbol{x}^{*(k)}) \quad \forall i \in \{1, ..., m\}. \tag{24}$$

If we replace the solution $\boldsymbol{x}^{*(k)}$ in $\boldsymbol{X}_K^*$ by $\hat{\boldsymbol{x}}$ (e.g., $\hat{\boldsymbol{X}}_K = (\boldsymbol{X}_K^* \setminus \{\boldsymbol{x}^{*(k)}\}) \cup \{\hat{\boldsymbol{x}}\}$), we have the new solution set $\hat{\boldsymbol{X}}_K = \{\boldsymbol{x}^{*(1)}, \ldots, \boldsymbol{x}^{*(k-1)}, \hat{\boldsymbol{x}}, \boldsymbol{x}^{*(k+1)}, \ldots, \boldsymbol{x}^{*(K)}\}$. Based on the above inequalities, and further let $\hat{\boldsymbol{x}}^{*(j)} = \boldsymbol{x}^{*(j)}, \forall j \neq k$ and $\hat{\boldsymbol{x}}^{*(k)} = \hat{\boldsymbol{x}}$, it is easy to check:

$$\mu \log \left( \sum_{i=1}^m e^{\frac{-\mu_i \log\left(\sum_{k=1}^K e^{-\lambda_i(f_i(\hat{\boldsymbol{x}}^{*(k)}) - z_i^*)/\mu_i}\right)}{\mu}} \right) < \mu \log \left( \sum_{i=1}^m e^{\frac{-\mu_i \log\left(\sum_{k=1}^K e^{-\lambda_i(f_i(\boldsymbol{x}^{*(k)}) - z_i^*)/\mu_i}\right)}{\mu}} \right). \tag{25}$$

There is a contradiction between (25) and the optimality of $\boldsymbol{X}_K^*$ for the smooth Tchebycheff set scalarization (23). Therefore, every $\boldsymbol{x}^{*(k)}$ should be a weakly Pareto optimal solution for the original multi-objective optimization problem (1).

Then we prove the two sufficient conditions for all solutions in $\boldsymbol{X}_K^*$ be Pareto optimal.

**1. Unique Optimal Solution Set** Without loss of generality, we suppose the $k$-th solution $\boldsymbol{x}^{*(k)}$ in $\boldsymbol{X}_K^*$ is not Pareto optimal, which means there exists a valid solution $\hat{\boldsymbol{x}} \in \mathcal{X}$ such that $\boldsymbol{f}(\hat{\boldsymbol{x}}) \prec \boldsymbol{f}(\boldsymbol{x}^{*(k)})$. In other words, we have:

$$f_i(\hat{\boldsymbol{x}}) \leq f_i(\boldsymbol{x}^{*(k)}), \forall i \in \{1, ..., m\} \text{ and } f_j(\hat{\boldsymbol{x}}) < f_j(\boldsymbol{x}^{*(k)}), \exists j \in \{1, ..., m\}. \tag{26}$$

Also let $\hat{\boldsymbol{X}}_K = (\boldsymbol{X}_K^* \setminus \{\boldsymbol{x}^{*(k)}\}) \cup \{\hat{\boldsymbol{x}}\}$ where $\hat{\boldsymbol{x}}^{*(j)} = \boldsymbol{x}^{*(j)}, \forall j \neq k$ and $\hat{\boldsymbol{x}}^{*(k)} = \hat{\boldsymbol{x}}$, we have:

$$\mu \log \left( \sum_{i=1}^m e^{\frac{-\mu_i \log\left( \sum_{k=1}^K e^{-\lambda_i(f_i(\hat{\boldsymbol{x}}^{*(k)}) - z_i^*)/\mu_i} \right)}{\mu}} \right) \leq \mu \log \left( \sum_{i=1}^m e^{\frac{-\mu_i \log\left( \sum_{k=1}^K e^{-\lambda_i(f_i(\boldsymbol{x}^{*(k)}) - z_i^*)/\mu_i} \right)}{\mu}} \right). \tag{27}$$

On the other hand, according to the uniqueness of $\boldsymbol{X}_K^*$, we have:

$$\mu \log \left( \sum_{i=1}^m e^{\frac{-\mu_i \log\left( \sum_{k=1}^K e^{-\lambda_i(f_i(\hat{\boldsymbol{x}}^{*(k)}) - z_i^*)/\mu_i} \right)}{\mu}} \right) > \mu \log \left( \sum_{i=1}^m e^{\frac{-\mu_i \log\left( \sum_{k=1}^K e^{-\lambda_i(f_i(\boldsymbol{x}^{*(k)}) - z_i^*)/\mu_i} \right)}{\mu}} \right). \tag{28}$$

The above two inequalities (27) and (28) are contradicted with each other. Therefore, every solution in the unique optimal solution set $\boldsymbol{x}^{*(k)}$ should be Pareto optimal.

**2. All Positive Preferences** Similar to the above proof, suppose the solution $\boldsymbol{x}^{*(k)}$ is not Pareto optimal, and there exists a valid solution $\hat{\boldsymbol{x}} \in \mathcal{X}$ such that $\boldsymbol{f}(\hat{\boldsymbol{x}}) \prec \boldsymbol{f}(\boldsymbol{x}^{*(k)})$. Since all preferences $\boldsymbol{\lambda} = \{\lambda_i\}_{i=1}^m$ are positive, according to the set of inequalities in (26), it is easy to check:

$$\mu \log \left( \sum_{i=1}^m e^{\frac{-\mu_i \log\left( \sum_{k=1}^K e^{-\lambda_i(f_i(\hat{\boldsymbol{x}}^{*(k)}) - z_i^*)/\mu_i} \right)}{\mu}} \right) < \mu \log \left( \sum_{i=1}^m e^{\frac{-\mu_i \log\left( \sum_{k=1}^K e^{-\lambda_i(f_i(\boldsymbol{x}^{*(k)}) - z_i^*)/\mu_i} \right)}{\mu}} \right) \tag{29}$$

which contradicts the STCH-Set optimality of $\boldsymbol{X}_K^*$ in (23). Therefore, all solutions in $\boldsymbol{X}_K^*$ should be Pareto optimal.

It should be noted that all positive preferences are not a sufficient conditions for the solution to be Pareto optimal for the original (nonsmooth) Tchebycheff set (TCH-Set) scalarization. With all positive preferences, each component (e.g., all objective-solution pairs) in STCH-Set scalarization will contribute to the scalarization value, which is not the case for the (non-smooth) TCH-Set scalarization.

$\square$

### A.3 Proof of Theorem 3

**Theorem 3** (Uniform Smooth Approximation). *The smooth Tchebycheff set (STCH-Set) scalarization $g_{\mu,\{\mu_i\}_{i=1}^m}^{(STCH\text{-}Set)}(\boldsymbol{X}_K|\boldsymbol{\lambda})$ is a uniform smooth approximation of the Tchebycheff set (TCH-Set) scalarization $g^{(TCH\text{-}Set)}(\boldsymbol{X}_K|\boldsymbol{\lambda})$, and we have:*

$$\lim_{\mu \downarrow 0, \mu_i \downarrow 0 \ \forall i} g_{\mu,\{\mu_i\}_{i=1}^m}^{(STCH\text{-}Set)}(\boldsymbol{X}_K|\boldsymbol{\lambda}) = g^{(TCH\text{-}Set)}(\boldsymbol{X}_K|\boldsymbol{\lambda}) \tag{30}$$

*for any valid set $\boldsymbol{X}_k \subset \mathcal{X}$.*

*Proof.* This theorem can be proved by deriving the upper and lower bounds of $g^{(\text{TCH-Set})}(\boldsymbol{X}_K|\boldsymbol{\lambda})$ with respect to the smooth smax and smin operators in $g_{\mu,\{\mu_i\}_{i=1}^m}^{(\text{STCH-Set})}(\boldsymbol{X}_K|\boldsymbol{\lambda})$. We first provide the upper and lower bounds for the classic smooth approximation for max and min, and then derive the bounds for $g^{(\text{TCH-Set})}(\boldsymbol{X}_K|\boldsymbol{\lambda})$.

**Classic Bounds for** max **and** min The log-sum-exp function $\log \sum_{i=1}^n e^{y_i}$ is a widely-used smooth approximation for the maximization function $\max\{y_1, y_2, \ldots, y_n\}$. Its lower and upper bounds are also well-known in convex optimization (Bertsekas et al., 2003; Boyd & Vandenberghe, 2004):

$$\max\{y_1, y_2, \ldots, y_n\} = \log(e^{\max\{y_1, y_2, \ldots, y_n\}})$$

$$\leq \log(\sum_{i=1}^{n} e^{y_i})$$

$$\leq \log(n \cdot e^{\max\{y_1, y_2, \dots, y_n\}})$$

$$= \log n + \log(e^{\max\{y_1, y_2, \dots, y_n\}})$$

$$= \log n + e^{\max\{y_1, y_2, \dots, y_n\}}. \tag{31}$$

By rearranging the above inequalities, we have:

$$\log \sum_{i=1}^{m} e^{y_i} - \log n \leq \max\{y_1, y_2, \dots, y_n\} \leq \log \sum_{i=1}^{m} e^{y_i}. \tag{32}$$

With a smooth parameter $\mu > 0$, it is also easy to show (Bertsekas et al., 2003; Boyd & Vandenberghe, 2004):

$$\mu \log \sum_{i=1}^{m} e^{y_i/\mu} - \mu \log n \leq \max\{y_1, y_2, \dots, y_n\} \leq \mu \log \sum_{i=1}^{m} e^{y_i/\mu}. \tag{33}$$

Similarly, we have the lower and upper bounds for the minimization function:

$$-\mu \log \sum_{i=1}^{m} e^{-y_i/\mu} \leq \min\{y_1, y_2, \dots, y_n\} \leq -\mu \log \sum_{i=1}^{m} e^{-y_i/\mu} + \mu \log n. \tag{34}$$

**Lower and Upper Bounds for $g^{(\text{TCH-Set})}(\boldsymbol{X}_K | \boldsymbol{\lambda})$** By leveraging the above classic bounds, it is straightforward to derive the lower and upper bounds for the $\max$ operator over $m$ optimization objectives for $g^{(\text{TCH-Set})}(\boldsymbol{X}_K | \boldsymbol{\lambda}) = \max_{1 \leq i \leq m} \left\{ \lambda_i \left( \min_{1 \leq k \leq K} f_i(\boldsymbol{x}^{(k)}) - z_i^* \right) \right\}$:

$$\max_{1 \leq i \leq m} \left\{ \lambda_i \left( \min_{1 \leq k \leq K} f_i(\boldsymbol{x}^{(k)}) - z_i^* \right) \right\} \geq \mu \log \left( \sum_{i=1}^{m} e^{\lambda_i \left( \min_{1 \leq k \leq K} f_i(\boldsymbol{x}^{(k)}) - z_i^* \right)/\mu} \right) - \mu \log m, \tag{35}$$

$$\max_{1 \leq i \leq m} \left\{ \lambda_i \left( \min_{1 \leq k \leq K} f_i(\boldsymbol{x}^{(k)}) - z_i^* \right) \right\} \leq \mu \log \left( \sum_{i=1}^{m} e^{\lambda_i \left( \min_{1 \leq k \leq K} f_i(\boldsymbol{x}^{(k)}) - z_i^* \right)/\mu} \right), \tag{36}$$

where the bounds are tight if $\mu \downarrow 0$.

In a similar manner, we can obtain the bounds for the $\min$ operator over $K$ solutions for each objective function $f_i(\boldsymbol{x})$:

$$-\mu_i \log \left( \sum_{k=1}^{K} e^{-f_i(\boldsymbol{x}^{(k)})/\mu_i} \right) \leq \min_{1 \leq k \leq K} f_i(\boldsymbol{x}^{(k)}) \leq -\mu_i \log \left( \sum_{k=1}^{K} e^{-f_i(\boldsymbol{x}^{(k)})/\mu_i} \right) + \mu_i \log K, \tag{37}$$

where the bounds are tight if $\mu_i \downarrow 0$.

Combing the results above, we have the upper bound and lower bounds for the Tchebycheff setscalarization $g^{(\text{TCH-Set})}(\boldsymbol{X}_K | \boldsymbol{\lambda})$:

$$g^{(\text{TCH-Set})}(\boldsymbol{X}_K | \boldsymbol{\lambda}) \geq \mu \log \left( \sum_{i=1}^{m} e^{\lambda_i \left( -\mu_i \log \left( \sum_{k=1}^{K} e^{-f_i(\boldsymbol{x}^{(k)})/\mu_i} \right) - z_i^* \right)/\mu} \right) - \mu \log m, \tag{38}$$

$$g^{(\text{TCH-Set})}(\boldsymbol{X}_K | \boldsymbol{\lambda}) \leq \mu \log \left( \sum_{i=1}^{m} e^{\lambda_i \left( -\mu_i \log \left( \sum_{k=1}^{K} e^{-f_i(\boldsymbol{x}^{(k)})/\mu_i} \right) + \mu_i \log K - z_i^* \right)/\mu} \right), \tag{39}$$

which means $g^{(\text{TCH-Set})}(\boldsymbol{X}_K | \boldsymbol{\lambda})$ is properly bounded from above and below, and the bounds are tight if $\mu \downarrow 0$ and $\mu_i \downarrow 0$ for all $1 \leq i \leq m$. It is straightforward to see:

$$\lim_{\mu \downarrow 0, \mu_i \downarrow 0 \; \forall i} g^{(\text{STCH-Set})}_{\mu, \{\mu_i\}_{i=1}^{m}}(\boldsymbol{X}_K | \boldsymbol{\lambda}) = g^{(\text{TCH-Set})}(\boldsymbol{X}_K | \boldsymbol{\lambda}) \tag{40}$$

for all valid solution sets $\boldsymbol{X}_K$ that include the optimal set $\boldsymbol{X}_K^* = \arg\min_{\boldsymbol{X}_K} g^{(\text{TCH-Set})}(\boldsymbol{X}_K | \boldsymbol{\lambda})$. Therefore, according to Nesterov (2005) (Nesterov, 2005), $g^{(\text{STCH-Set})}_{\mu, \{\mu_i\}_{i=1}^{m}}$ is a uniform smooth approximation of $g^{(\text{TCH-Set})}(\boldsymbol{X}_K | \boldsymbol{\lambda})$.

$\square$

## A.4 PROOF OF THEOREM 4

**Theorem 4** (Convergence to Pareto Stationary Solution). *If there exists a solution set $\hat{X}_K$ such that $\nabla_{\hat{x}^{(k)}} g_{\mu,\{\mu_i\}_{i=1}^m}^{(STCH\text{-}Set)}(\hat{X}_K|\lambda) = 0$ for all $\hat{x}^{(k)} \in \hat{X}_K$, then all solutions in $\hat{X}_K$ are Pareto stationary solutions of the original multi-objective optimization problem (1).*

*Proof.* We can prove this theorem by analyzing the form of gradient for STCH-Set scalarization $\nabla_{\hat{x}^{(k)}} g_{\mu,\{\mu_i\}_{i=1}^m}^{(STCH\text{-}Set)}(\hat{X}_K|\lambda) = 0$ for each solution $x^{(k)}$ with the condition for Pareto stationarity in Definition 4.

We first let

$$g_i(\boldsymbol{x}) = -\mu_i \log\left(\sum_{k=1}^K e^{-f_i(\boldsymbol{x}^{(k)})/\mu_i}\right) \quad \forall 1 \le i \le m, \tag{41}$$

$$\boldsymbol{y} = \boldsymbol{\lambda}(\boldsymbol{g}(\boldsymbol{x}) - \boldsymbol{z}^*) \in \mathbb{R}^m, \tag{42}$$

$$h(\boldsymbol{y}) = -\mu \log\left(\sum_{i=1}^m e^{-\boldsymbol{y}_i/\mu}\right). \tag{43}$$

For the STCH scalarization, we have:

$$g_{\mu,\{\mu_i\}_{i=1}^m}^{(STCH\text{-}Set)}(\boldsymbol{X}_K|\boldsymbol{\lambda}) = \mu \log\left(\sum_{i=1}^m e^{\frac{\lambda_i\left(-\mu_i \log\left(\sum_{k=1}^K e^{-f_i(\boldsymbol{x}^{(k)})/\mu_i}\right) - z_i^*\right)}{\mu}}\right) \tag{44}$$

$$= \mu \log\left(\sum_{i=1}^m e^{\frac{\lambda_i(g_i(\boldsymbol{x}) - z_i^*)}{\mu}}\right) \tag{45}$$

$$= \mu \log\left(\sum_{i=1}^m e^{\frac{y_i}{\mu}}\right) \tag{46}$$

$$= h(\boldsymbol{y}). \tag{47}$$

Therefore, according to the chain rule, the gradient of $g_{\mu,\{\mu_i\}_{i=1}^m}^{(STCH\text{-}Set)}(\boldsymbol{X}_K|\boldsymbol{\lambda})$ with respect to a solution $\boldsymbol{x}^{(k)}$ in the set $\boldsymbol{X}_K$ can be written as:

$$\nabla_{\boldsymbol{x}^{(k)}} g_{\mu,\{\mu_i\}_{i=1}^m}^{(STCH\text{-}Set)}(\boldsymbol{X}_K|\boldsymbol{\lambda}) = \nabla_{\boldsymbol{y}} h(\boldsymbol{y}) \cdot \frac{\partial \boldsymbol{y}}{\partial \boldsymbol{g}} \cdot \frac{\partial \boldsymbol{g}}{\partial \boldsymbol{x}^{(k)}}. \tag{48}$$

It is straightforward to show

$$\nabla_{\boldsymbol{y}} h(\boldsymbol{y}) = \frac{e^{\boldsymbol{y}/\mu}}{\sum_i e^{y_i/\mu}}, \quad \frac{\partial \boldsymbol{y}}{\partial \boldsymbol{g}} = \boldsymbol{\lambda}, \quad \frac{\partial \boldsymbol{g}_i}{\partial \boldsymbol{x}^{(k)}} = \frac{e^{-f_i(\boldsymbol{x}^{(k)})/\mu_i}}{\sum_k e^{-f_i(\boldsymbol{x}^{(k)})/\mu_i}}. \tag{49}$$

Therefore, we have

$$\nabla_{\boldsymbol{x}^{(k)}} g_{\mu,\{\mu_i\}_{i=1}^m}^{(STCH\text{-}Set)}(\boldsymbol{X}_K|\boldsymbol{\lambda}) = \sum_{i=1}^m \frac{\lambda_i e^{y_i/\mu}}{\sum_i e^{y_i/\mu}} \frac{e^{-f_i(\boldsymbol{x}^{(k)})/\mu_i}}{\sum_k e^{-f_i(\boldsymbol{x}^{(k)})/\mu_i}} \nabla f_i(\boldsymbol{x}^{(k)}). \tag{50}$$

Let $w_i = \frac{\lambda_i e^{y_i/\mu}}{\sum_i e^{y_i/\mu}} \frac{e^{-f_i(\boldsymbol{x}^{(k)})/\mu_i}}{\sum_k e^{-f_i(\boldsymbol{x}^{(k)})/\mu_i}}$, it is easy to check $w_i \ge 0 \; \forall 1 \le i \le m$. If set $s = \sum_{i=1}^m w_i > 0$ and $\bar{w}_i = w_i/s$, we have:

$$\nabla_{\boldsymbol{x}^{(k)}} g_{\mu,\{\mu_i\}_{i=1}^m}^{(STCH\text{-}Set)}(\boldsymbol{X}_K|\boldsymbol{\lambda}) = \sum_{i=1}^m \frac{\lambda_i e^{y_i/\mu}}{\sum_i e^{y_i/\mu}} \frac{e^{-f_i(\boldsymbol{x}^{(k)})/\mu_i}}{\sum_k e^{-f_i(\boldsymbol{x}^{(k)})/\mu_i}} \nabla f_i(\boldsymbol{x}^{(k)}), \tag{51}$$

$$= \sum_{i=1}^m w_i \nabla f_i(\boldsymbol{x}) = s \sum_{i=1}^m \bar{w}_i \nabla f_i(\boldsymbol{x}) \tag{52}$$

$$\propto \sum_{i=1}^m \bar{w}_i \nabla f_i(\boldsymbol{x}), \tag{53}$$

where $\bar{\boldsymbol{w}} \in \boldsymbol{\Delta}^{m-1} = \{\bar{\boldsymbol{w}}| \sum_{i=1}^{m} \bar{w}_i = 1, \bar{w}_i \geq 0 \ \forall i\}$. For a solution $\hat{\boldsymbol{x}}^{(k)} \in \hat{\boldsymbol{X}}_K$, if $\nabla_{\hat{\boldsymbol{x}}^{(k)}} g_{\mu,\{\mu_i\}_{i=1}^{m}}^{\text{(STCH-Set)}}(\hat{\boldsymbol{X}}_K|\boldsymbol{\lambda}) = \boldsymbol{0}$, we have:

$$\sum_{i=1}^{m} \bar{w}_i \nabla f_i(\boldsymbol{x}) = \boldsymbol{0} \text{ with } \bar{\boldsymbol{w}} \in \boldsymbol{\Delta}^{m-1} = \{\bar{\boldsymbol{w}}| \sum_{i=1}^{m} \bar{w}_i = 1, \bar{w}_i \geq 0 \ \forall i\}. \tag{54}$$

According to Definition 4, the solution $\boldsymbol{x}^{(k)}$ is Pareto stationary for the multi-objective optimization problem (1).

Therefore, when $\nabla_{\hat{\boldsymbol{x}}^{(k)}} g_{\mu,\{\mu_i\}_{i=1}^{m}}^{\text{(STCH-Set)}}(\hat{\boldsymbol{X}}_K|\boldsymbol{\lambda}) = \boldsymbol{0}$ for all $\hat{\boldsymbol{x}}^{(k)} \in \hat{\boldsymbol{X}}_K$, all solutions in $\hat{\boldsymbol{X}}_K$ are Pareto stationary for the original multi-objective optimization problem (1).

$\square$

### A.5 DISCUSSION ON THE PARETO OPTIMALITY GUARANTEE FOR (S)TCH-SET

According to **Theorem 1**, without the strong unique solution set assumption, the solutions in a general optimal solution set of TCH-Set are not necessarily weakly Pareto optimal. This result is a bit surprising yet reasonable for the set-based few-for-many problem. The key reason here is that only the single worst weighted objective (and its corresponding solution) will contribute the TCH-Set scalarization value $\max_{1 \leq i \leq m}\{\lambda_i(\min_{\boldsymbol{x} \in \boldsymbol{X}_K} f_i(\boldsymbol{x}) - \boldsymbol{z}_i^*)\}$. In other words, the rest of the solutions can have arbitrary performance and will not affect the TCH-Set scalarization value. For the classic multi-objective optimization problem (e.g., $K = 1$), this property makes the TCH scalarization only have a weakly Pareto optimality guarantee (only the worst objective counts), and which becomes worse for the few-for-many setting.

**Counterexample** We provide a counterexample to better illustrate this not-even-weakly-Pareto-optimal property. Consider a 2-solution-for-3-objective problem where the best values for the three objectives are $(1, 1, 1)$. With preference $(0.98, 0.01, 0.01)$, an ideal optimal solution set can be two weakly Pareto optimal solutions with values $\{(1, 8, 8), (3, 1, 1)\}$. However, since only the largest weighted objective $\lambda_1 f_1(x_1) = 0.98$ will contribute to the final TCH-Set value, we can freely make the other solution have a worse objective value. For example, the set of solutions with value $\{(1, 8, 8), (3, 2, 2)\}$ is still the optimal solution set for TCH-Set, but the second solution is clearly not weakly Pareto optimal. Neither the positive preference assumption nor the no redundant solution assumption can exclude this kind of counterexample.

Table 4: (Weakly) Pareto optimaility guarantee for TCH-Set and STCH-Set.

| Assumption | - | All Positive Preferences | Unique Solution Set |
|---|---|---|---|
| TCH-Set | - | - | Pareto Optimal |
| STCH-Set | Weakly Pareto Optimal | Pareto Optimal | Pareto Optimal |

**(Weakly) Pareto optimality guarantee for STCH-Set** In contrast to TCH-Set, STCH-Set enjoys a good (weakly) Pareto optimality guarantee for its optimal solution set. The key reason is that all objectives and solutions will contribute to the STCH-Set scalarization value. According to **Theorem 2**, all optimal solutions for STCH-Set are at least weakly Pareto optimal, and they are Pareto optimal if 1) all preferences are positive or 2) the optimal solution set is unique. The comparison of TCH-Set and STCH-Set with different assumptions can be found in Table 4.

## B PROBLEM AND EXPERIMENTAL SETTINGS

### B.1 CONVEX MULTI-OBJECTIVE OPTIMIZATION

In this experiment, we compare different methods on solving $m$ convex optimization problems with $K$ solutions. We consider two numbers of objectives $m = \{128, 1024\}$ and six different numbers of solutions $K = \{3, 4, 5, 6, 8, 10\}$ or $K = \{3, 5, 8, 10, 15, 20\}$ for the 128-objective and 1024-objective problems separately. Therefore, there are total $2 \times 6 = 12$ comparisons.

For each comparison, we randomly generate $m$ independent quadratic functions $\{f_i(\boldsymbol{x})\}_{i=1}^m$ as the optimization objectives, of which the minimum values are all 0. Therefore, the optimal worst and average objective values are both 0 for all comparisons. Since the $m$ objectives are randomly generated and with different optimal solution, the optimal worst and average objective value 0 cannot be achieved by any algorithms unless $K \geq m$. In all comparison, we let all solution be 10-dimensional vector $\boldsymbol{x} \in \mathbb{R}^{10}$. We repeat each comparison 50 times and report the mean worst and average objective value over 50 runs for each method.

### B.2 NOISY MIXED LINEAR REGRESSION

We follow the noisy mixed linear regression setting from (Ding et al., 2024). The $i$-th optimization objective is defined as:

$$f_i(\boldsymbol{x}) = \frac{1}{2}(\boldsymbol{a}^{(i)T}\boldsymbol{x} - b^{(i)})^2 + \frac{\beta}{2}||\boldsymbol{x}||^2 \tag{55}$$

where $\{(\boldsymbol{a}^{(i)}, b^{(i)})\}_{i=1}^m$ are $m$ data points and $\beta = 0.01$ is a fixed penalty parameter. The dataset $\{(\boldsymbol{a}^{(i)}, b^{(i)})\}_{i=1}^m$ is randomly generated in the following steps:

- Randomly sample $K$ i.i.d. ground truth $\{\hat{\boldsymbol{x}}^{(k)}\}_{k=1}^K \sim N(\boldsymbol{0}, \boldsymbol{I}_d)$;
- Randomly sample $m$ i.i.d. data $\{\boldsymbol{a}^{(i)}\}_{i=1}^m \sim N(\boldsymbol{0}, \boldsymbol{I}_d)$, class index $\{c_i\}_{i=1}^m \sim \text{Uniform}([k])$, and noise $\{\epsilon_i\}_{i=1}^m \sim N(\boldsymbol{0}, \sigma^2)$;
- Compute $b^{(i)} = \boldsymbol{a}^{(i)T}\hat{\boldsymbol{x}}^{(c_i)} + \epsilon_i$.

Our goal is to build $K$ linear model with parameter $\boldsymbol{x}^{(k)}$ to tackle all $m$ data points. In this experiment, we set $m = 1000$ and $\boldsymbol{x} \in \mathbb{R}^{10}$ for all comparisons. We consider three noise levels $\sigma = \{0.1, 0.5, 1\}$ and four different numbers of solutions $K = \{5, 10, 15, 20\}$ for each noise level, so there are total $3 \times 4 = 12$ comparisons. All comparisons are run 50 times, and we compare the mean worst and average objective values for each method.

### B.3 NOISY MIXED NONLINEAR REGRESSION

The experimental setting for noisy mixed nonlinear regression is also from (Ding et al., 2024) and similar to the linear regression counterpart. For nonlinear regression, the $i$-th optimization objective is defined as:

$$f_i(\boldsymbol{x}) = \frac{1}{2}(\psi(\boldsymbol{a}_i; \boldsymbol{x}) - b_i)^2 + \frac{\beta}{2}||\boldsymbol{x}||^2 \tag{56}$$

where $\{(a_i, b_i)\}_{i=1}^m$ are $m$ data points and $\beta = 0.01$ is a fixed penalty parameter as in the previous linear regression case. However, now the model $\psi(\boldsymbol{a}_i, \boldsymbol{x})$ is a neural network with trainale model parameters $\boldsymbol{x} = (\boldsymbol{W}, \boldsymbol{p}, \boldsymbol{q}, o)$ with the form:

$$\psi(\boldsymbol{a}; \boldsymbol{x}) = \psi(\boldsymbol{a}; \boldsymbol{W}, \boldsymbol{p}, \boldsymbol{q}, o) = \boldsymbol{p}^T \text{ReLU}(\boldsymbol{W}\boldsymbol{a} + q) + o, \tag{57}$$

where $\boldsymbol{a} \in \mathbb{R}^{d_I}, \boldsymbol{W} \in \mathbb{R}^{d_H \times d_I}, \boldsymbol{p}, \boldsymbol{q} \in \mathbb{R}^{d_H}$ and $o \in \mathbb{R}$. The $d_I$ and $d_H$ are the input dimension and hidden dimension, respectively. The data set $\{(\boldsymbol{a}^{(i)}, b^{(i)})\}_{i=1}^m$ can be generated similar to linear regression but with $K$ ground truth neural network models. We set $d_I = 10$ and $d_H = 10$ in all comparisons.

## B.4 DEEP MULTI-TASK GROUPING

In this experiment, we follow the same setting for the CelebA with 9 tasks in Gao et al. (2024). The goal is to build a few multi-task learning models ($K = 2, 3$ or $4$) to handle all $m = 9$ tasks, where the optimal task group assignment is unknown in advance. The 9 tasks are all classification problems (e.g., 5-o-Clock Shadow, Black Hair, Blond Hair, Brown Hair, Goatee, Mustache, No Beard, Rosy Cheeks, and Wearing Hat) which are first used in Fifty et al. (2021) for multi-task grouping. This is a typical few-for-many optimization problem.

Following the setting in Gao et al. (2024), for all methods, we use a ResNet variant as the network backbone and the cross-entropy loss for all tasks (the same setting as in Fifty et al. (2021)). All models are trained by the Adam optimizer with initial learning rates $0.0008$ with plateau learning rate decay for 100 epochs. The results of the existing multi-task grouping methods (Random Grouping, HOA (Standley et al., 2020), TAG (Fifty et al., 2021) and DMTG (Gao et al., 2024)) are directly from Gao et al. (2024). We train the models with SoM, TCH-Set and STCH-Set ourselves using the codes provided by Gao et al. (2024) [2]. More experimental details for this problem can be found in Gao et al. (2024).

---

[2]https://github.com/ethanygao/DMTG

## C MORE EXPERIMENTAL RESULTS

### C.1 CONVEX MANY-OBJECTIVE OPTIMIZATION

Table 5: The results on the convex optimization problems with different numbers of solutions $K$. Mean worst and average objective values over $50$ runs are reported. Best results are highlighted in **bold** with gray background.

| | | LS | TCH | STCH | MosT | SoM | TCH-Set | STCH-Set |
|---|---|---|---|---|---|---|---|---|
| | | | | number of objectives $m = 128$ | | | | |
| $K = 3$ | worst | 4.64e+00(+) | 4.44e+00(+) | 4.41e+00(+) | 2.12e+00(+) | 1.86e+00(+) | 1.02e+00(+) | **6.08e-01** |
| | average | 8.61e-01(+) | 8.03e-01(+) | 7.80e-01(+) | 3.45e-01(+) | **2.02e-01(-)** | 3.46e-01(+) | 2.12e-01 |
| $K = 4$ | worst | 4.23e+00(+) | 3.83e+00(+) | 3.68e+00(+) | 1.48e+00(+) | 1.12e+00(+) | 6.74e-01(+) | **3.13e-01** |
| | average | 7.45e-01(+) | 6.54e-01(+) | 6.41e-01(+) | 1.85e-01(+) | 1.12e-01(+) | 2.27e-01(+) | **9.44e-02** |
| $K = 5$ | worst | 4.17e+00(+) | 3.56e+00(+) | 3.51e+00(+) | 1.02e+00(+) | 9.20e-01(+) | 4.94e-01(+) | **1.91e-01** |
| | average | 7.08e-01(+) | 5.75e-01(+) | 5.05e-01(+) | 9.81e-02(+) | 7.95e-02(+) | 1.60e-01(+) | **5.12e-02** |
| $K = 6$ | worst | 3.81e+00(+) | 3.41e+00(+) | 3.20e+00(+) | 9.56e-01(+) | 7.03e-01(+) | 3.72e-01(+) | **1.36e-01** |
| | average | 6.72e-01(+) | 5.31e-01(+) | 5.15e-01(+) | 8.34e-02(+) | 5.34e-02(+) | 1.19e-01(+) | **3.15e-02** |
| $K = 8$ | worst | 3.69e+00(+) | 2.94e+00(+) | 2.61e+00(+) | 8.32e-01(+) | 5.20e-01(+) | 2.84e-01(+) | **1.02e-01** |
| | average | 6.16e-01(+) | 4.57e-01(+) | 4.22e-01(+) | 6.91e-02(+) | 3.40e-02(+) | 8.53e-02(+) | **1.78e-02** |
| $K = 10$ | worst | 3.68e+00(+) | 2.69e+00(+) | 2.40e+00(+) | 6.27e-01(+) | 4.07e-01(+) | 1.95e-01(+) | **7.99e-02** |
| | average | 5.69e-01(+) | 4.07e-01(+) | 3.20e-01(+) | 4.59e-02(+) | 2.43e-02(+) | 5.92e-02(+) | **1.34e-02** |
| | | | | number of objectives $m = 1024$ | | | | |
| $K = 3$ | worst | 4.71e+00(+) | 7.84e+00(+) | 7.33e+00(+) | - | 3.36e+00(+) | 2.39e+00(+) | **1.87e+00** |
| | average | 1.15e+00(+) | 9.92e-01(+) | 9.70e-01(+) | - | **3.54e-01(-)** | 4.55e-01(+) | 3.92e-01 |
| $K = 5$ | worst | 4.48e+00(+) | 6.77e+00(+) | 6.57e+00(+) | - | 2.34e+00(+) | 1.68e+00(+) | **9.93e-01** |
| | average | 1.09e+00(+) | 8.02e-01(+) | 7.65e-01(+) | - | 1.95e-01(+) | 2.59e-01(+) | **1.87e-01** |
| $K = 8$ | worst | 4.25e+00(+) | 5.65e+00(+) | 5.58e+00(+) | - | 1.74e+00(+) | 1.25e+00(+) | **4.90e-01** |
| | average | 1.03e+00(+) | 6.69e-01(+) | 6.49e-01(+) | - | 1.02e-01(+) | 1.66e-01(+) | **8.31e-02** |
| $K = 10$ | worst | 4.16e+00(+) | 5.61e+00(+) | 5.44e+00(+) | - | 1.78e+00(+) | 1.16e+00(+) | **4.05e-01** |
| | average | 1.00e+00(+) | 6.07e-01(+) | 5.99e-01(+) | - | 8.14e-02(+) | 1.41e-01(+) | **5.94e-02** |
| $K = 15$ | worst | 4.06e+00(+) | 4.78e+00(+) | 4.91e+00(+) | - | 1.54e+00(+) | 9.06e-01(+) | **2.85e-01** |
| | average | 9.78e-01(+) | 5.15e-01(+) | 4.99e-01(+) | - | 5.98e-02(+) | 1.01e-01(+) | **3.31e-02** |
| $K = 20$ | worst | 3.97e+00(+) | 4.64e+00(+) | 4.57e+00(+) | - | 1.52e+00(+) | 8.04e-01(+) | **2.28e-01** |
| | average | 9.56e-01(+) | 4.63e-01(+) | 4.68e-01(+) | - | 5.30e-02(+) | 8.40e-02(+) | **2.27e-02** |
| | | | | Wilcoxon Rank-Sum Test Summary | | | | |
| $+/=/-$ | worst | 12/0/0 | 12/0/0 | 12/0/0 | 6/0/0 | 12/0/0 | 12/0/0 | - |
| | average | 12/0/0 | 12/0/0 | 12/0/0 | 6/0/0 | 10/0/2 | 12/0/0 | - |

The full results for the convex many-objective optimization are shown in Table 5. Since the MosT method cannot tackle the problems with $1024$ objectives in a reasonable time, we did not include these results in the Table 5 and marked it as "-". According to the results, STCH-Set can always achieve the best worst case performance for all comparisons and the best average performance for most cases.

## C.2 NOISY MIXED LINEAR REGRESSION

Table 6: The results on the noisy mixed linear regression with different different noisy levels $\sigma$ and different numbers of solutions $K$. We report the mean worst and average objective values over 50 independent runs for each method. The best results are highlighted in **bold** with the gray background.

| | | LS | TCH | STCH | SoM | TCH-Set | STCH-Set |
|---|---|---|---|---|---|---|---|
| | | | | $\sigma = 0.1$ | | | |
| $K = 5$ | worst | 3.84e+01(+) | 2.45e+01(+) | 2.43e+01(+) | 2.50e+00(+) | 4.19e+00(+) | **2.10e+00** |
| | average | 2.70e+00(+) | 1.46e+00(+) | 1.44e+00(+) | **1.88e-01(-)** | 6.53e-01(+) | 4.72e-01 |
| $K = 10$ | worst | 2.21e+01(+) | 2.49e+01(+) | 3.92e+01(+) | 3.46e+00(+) | 2.04e+00(+) | **5.00e-01** |
| | average | 1.09e+00(+) | 1.19e+00(+) | 3.01e+00(+) | 2.33e-01(+) | 3.83e-01(+) | **2.00e-01** |
| $K = 15$ | worst | 4.20e+01(+) | 2.11e+01(+) | 2.21e+01(+) | 2.57e+00(+) | 1.64e+00(+) | **2.70e-01** |
| | average | 3.07e+00(+) | 1.02e+00(+) | 1.02e+00(+) | 1.97e-01(+) | 3.22e-01(+) | **1.66e-01** |
| $K = 20$ | worst | 4.20e+01(+) | 2.14e+01(+) | 2.04e+01(+) | 1.44e+00(+) | 1.79e+00(+) | **2.27e-01** |
| | average | 3.09e+00(+) | 8.35e-01(+) | 8.87e-01(+) | 1.84e-01(+) | 3.29e-01(+) | **1.66e-01** |
| | | | | $\sigma = 0.5$ | | | |
| $K = 5$ | worst | 3.93e+01(+) | 2.53e+01(+) | 2.50e+01(+) | 5.33e+00(+) | 4.37e+00(+) | **2.45e+00** |
| | average | 2.74e+00(+) | 1.51e+00(+) | 1.52e+00(+) | **3.08e-01(-)** | 7.05e-01(+) | 5.54e-01 |
| $K = 10$ | worst | 4.20e+01(+) | 2.39e+01(+) | 2.56e+01(+) | 3.68e+00(+) | 2.38e+00(+) | **6.03e-01** |
| | average | 3.25e+00(+) | 1.28e+00(+) | 1.20e+00(+) | 2.46e-01(+) | 4.18e-01(+) | **2.24e-01** |
| $K = 15$ | worst | 4.36e+01(+) | 2.37e+01(+) | 2.35e+01(+) | 3.00e+00(+) | 1.72e+00(+) | **3.07e-01** |
| | average | 3.30e+00(+) | 1.02e+00(+) | 1.09e+00(+) | 2.06e-01(+) | 3.46e-01(+) | **1.78e-01** |
| $K = 20$ | worst | 4.36e+01(+) | 2.18e+01(+) | 2.23e+01(+) | 1.93e+00(+) | 1.27e+00(+) | **2.39e-01** |
| | average | 3.22e+00(+) | 9.15e-01(+) | 9.99e-01(+) | 1.92e-01(+) | 2.94e-01(+) | **1.74e-01** |
| | | | | $\sigma = 1$ | | | |
| $K = 5$ | worst | 4.76e+01(+) | 3.24e+01(+) | 3.48e+01(+) | 1.00e+01(+) | 5.49e+00(+) | **3.48e+00** |
| | average | 3.39e+00(+) | 1.71e+00(+) | 1.93e+00(+) | **5.21e-01(-)** | 8.45e-01(+) | 7.44e-01 |
| $K = 10$ | worst | 4.86e+01(+) | 2.93e+01(+) | 3.30e+01(+) | 4.86e+00(+) | 3.31e+00(+) | **7.82e-01** |
| | average | 3.81e+00(+) | 1.45e+00(+) | 1.55e+00(+) | 2.88e-01(+) | 5.30e-01(+) | **2.80e-01** |
| $K = 15$ | worst | 5.23e+01(+) | 3.11e+01(+) | 2.78e+01(+) | 3.41e+00(+) | 2.45e+00(+) | **3.55e-01** |
| | average | 3.84e+00(+) | 1.19e+00(+) | 1.16e+00(+) | 2.32e-01(+) | 4.51e-01(+) | **2.04e-01** |
| $K = 20$ | worst | 5.37e+01(+) | 2.54e+01(+) | 2.61e+01(+) | 2.47e+00(+) | 2.04e+00(+) | **2.68e-01** |
| | average | 3.81e+00(+) | 1.08e+00(+) | 1.08e+00(+) | 2.12e-01(+) | 3.81e-01(+) | **1.94e-01** |
| | | | | Wilcoxon Rank-Sum Test Summary | | | |
| $+/=/-$ | worst | 12/0/0 | 12/0/0 | 12/0/0 | 12/0/0 | 12/0/0 | - |
| | average | 12/0/0 | 12/0/0 | 12/0/0 | 12/0/0 | 9/0/3 | - |

The full results for the noisy mixed linear regression problem are shown in Table 6. According to the results, STCH-Set can always obtain the lowest worst objective values for all noise levels and different numbers of solutions. In addition, it also obtains the best average objective values for most comparisons. The sum-of-minimum (SoM) optimization method can achieve good and even better average objective values, but with a significantly higher worst objective value.

## C.3 NOISY MIXED NONLINEAR REGRESSION

Table 7: The results on the noisy mixed nonlinear regression with different noisy levels $\sigma$ and different numbers of solutions $K$. We report the mean worst and average objective values over 50 independent runs for each method. The best results are highlighted in **bold** with gray background.

| | | LS | TCH | STCH | SoM | TCH-Set | STCH-Set |
|---|---|---|---|---|---|---|---|
| | | | | $\sigma = 0.1$ | | | |
| $K = 5$ | worst | 2.34e+02 (+) | 1.68e+02(+) | 1.57e+02(+) | 1.68e+01(+) | 1.94e+01(+) | **7.43e+00** |
| | average | 9.19e+00(+) | 8.50e+00(+) | 7.90e+00(+) | **8.54e-01(-)** | 3.48e+00(+) | 1.89e+00 |
| $K = 10$ | worst | 3.23e+02(+) | 1.72e+02(+) | 1.57e+02(+) | 4.42e+00(+) | 6.07e+00(+) | **6.28e-01** |
| | average | 8.70e+00 (+) | 6.65e+00(+) | 5.85e+00(+) | 1.22e-01(+) | 1.03e+00(+) | **5.99e-02** |
| $K = 15$ | worst | 3.65e+02(+) | 2.10e+02(+) | 1.62e+02(+) | 1.10e+00(+) | 1.57e+00(+) | **2.05e-01** |
| | average | 8.33e+00(+) | 5.66e+00(+) | 4.89e+00(+) | 1.47e-01(+) | 3.27e-01(+) | **1.27e-02** |
| $K = 20$ | worst | 3.36e+02(+) | 2.04e+02(+) | 1.81e+02(+) | 5.59e+00(+) | 4.37e+00(+) | **6.22e-01** |
| | average | 8.81e+00(+) | 6.92e+00(+) | 6.23e+00(+) | 1.24e-01(+) | 9.09e-01(+) | **6.27e-02** |
| | | | | $\sigma = 0.5$ | | | |
| $K = 5$ | worst | 2.14e+02(+) | 1.87e+02(+) | 1.62e+02(+) | 2.00e+01(+) | 1.82e+01(+) | **1.20e+01** |
| | average | 9.17e+00(+) | 8.53e+00(+) | 7.96e+00(+) | **8.63e-01(-)** | 3.49e+00(+) | 2.31e+00 |
| $K = 10$ | worst | 3.36e+02(+) | 2.04e+02(+) | 1.81e+02(+) | 5.59e+00(+) | 4.37e+00(+) | **6.22e-01** |
| | average | 8.81e+00(+) | 6.92e+00(+) | 6.23e+00(+) | 1.24e-01(+) | 9.09e-01(+) | **6.27e-02** |
| $K = 15$ | worst | 3.46e+02(+) | 1.99e+02(+) | 1.53e+02(+) | 2.00e+00(+) | 2.15e+00(+) | **1.57e-01** |
| | average | 8.38e+00(+) | 5.63e+00(+) | 4.59e+00(+) | 1.81e-02(+) | 3.97e-01(+) | **1.26e-02** |
| $K = 20$ | worst | 3.75e+02(+) | 1.54e+02(+) | 1.27e+02(+) | 5.05e-01(+) | 6.60e-01(+) | **1.12e-01** |
| | average | 6.95e+00(+) | 4.23e+00(+) | 3.32e+00(+) | **3.96e-03(-)** | 1.35e-01(+) | 6.72e-03 |
| | | | | $\sigma = 1$ | | | |
| $K = 5$ | worst | 3.47e+02(+) | 1.50e+02(+) | 1.39e+02(+) | 1.81e-01(+) | 4.65e-01(+) | **9.49e-02** |
| | average | 6.95e+00(+) | 4.34e+00(+) | 3.32e+00(+) | **1.26e-03(-)** | 1.11e-01(+) | 6.65e-03 |
| $K = 10$ | worst | 3.08e+02(+) | 2.26e+02(+) | 1.68e+02(+) | 5.08e+00(+) | 6.91e+00(+) | **6.44e-01** |
| | average | 9.18e+00(+) | 7.10e+00(+) | 6.13e+00(+) | 1.32e-01(+) | 1.10e+00(+) | **5.77e-02** |
| $K = 15$ | worst | 3.35e+02(+) | 2.03e+02(+) | 1.64e+02(+) | 1.89e+00(+) | 1.91e+00(+) | **2.11e-01** |
| | average | 8.47e+00(+) | 6.19e+00(+) | 4.92e+00(+) | 2.11e-02(+) | 3.93e-01(+) | **1.30e-02** |
| $K = 20$ | worst | 3.40e+02(+) | 1.83e+02(+) | 1.21e+02(+) | 1.81e-01(+) | 4.65e-01(+) | **9.08e-02** |
| | average | 7.20e+00(+) | 4.44e+00(+) | 3.45e+00(+) | **3.24e-03(-)** | 1.11e-01(+) | 6.73e-03 |
| | | | | Wilcoxon Rank-Sum Test Summary | | | |
| $+/=/-$ | worst | 12/0/0 | 12/0/0 | 12/0/0 | 12/0/0 | 12/0/0 | - |
| | average | 12/0/0 | 12/0/0 | 12/0/0 | 7/0/5 | 12/0/0 | - |

The full results for the noisy mixed nonlinear regression problem are provided in Table 7. Similar to the linear counterpart, STCH-Set can also achieve the lowest worst objective value for all comparisons. However, it is outperformed by SoM on 5 out of 12 comparisons on the average objective value. One possible result could be due to the highly non-convex nature of neural network training. Once a solution is captured by a bad local optimum, it will have many high but not reducible objective values, which might mislead the STCH-Set optimization. An adaptive estimation method for the ideal point could be helpful to tackle this issue.

# D    ABLATION STUDIES AND DISCUSSION

## D.1    THE EFFECT OF SMOOTH PARAMETER

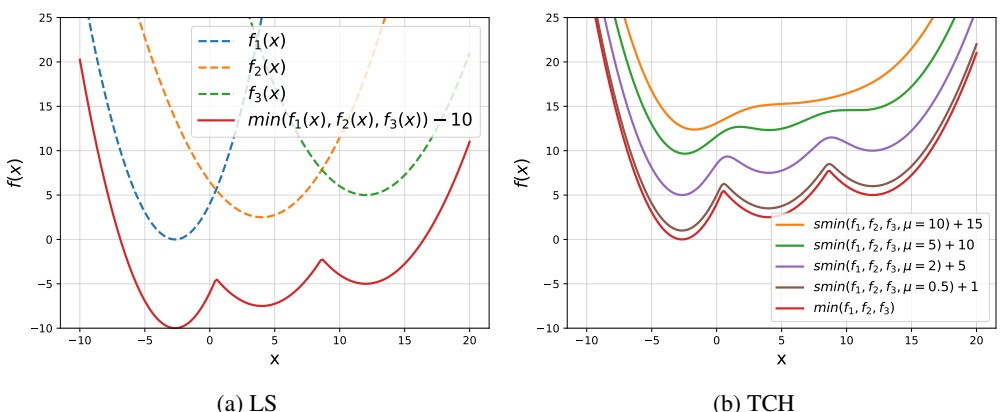

(a) LS                                                      (b) TCH

Figure 4: **The effect of different smoothing parameters $\mu$ for the smooth minimization function:**
(a) The function $\min\{f_1(x), f_2(x), f_3(x)\}$ is highly non-convex even when all $\{f_i(x)\}_{i=1}^3$ are convex.
(b) The smooth $\texttt{smin}$ function has a better optimization landscape, especially with large $\mu$. When
$\mu \to 0$, $\texttt{smin}$ converges to $\texttt{min}$.

Table 8: Results of STCH-Set with different fixed $\mu$ and an adaptive $\mu$ schedule.

|  |  | $\mu = 10$ | $\mu = 5$ | $\mu = 1$ | $\mu = 0.5$ | $\mu = 0.1$ | Adaptive $\mu$ |
|---|---|---|---|---|---|---|---|
| $K = 5$ | worst | 4.55e+00 | 4.38e+00 | 1.88e+00 | 1.37e+00 | 1.20e+00 | **9.93e-01** |
|  | average | 1.25e+00 | 1.23e+00 | 1.95e-01 | 1.99e-01 | 1.94e-01 | **1.87e-01** |
| $K = 8$ | worst | 4.40e+00 | 4.00e+00 | 1.39e+00 | 9.20e-01 | 6.62e-01 | **4.90e-01** |
|  | average | 1.19e+00 | 1.10e+00 | 9.44e-02 | 9.38e-02 | 9.06e-02 | **8.31e-02** |

In this work, we use the same smoothing parameters for all smooth terms (e.g., $\mu = \mu_1 = \mu_2, \ldots, \mu_m$
as in equation (13)) in STCH-Set for all experiments. The smoothing parameter is a hyperparameter
for STCH-Set, and might need to be tuned for different problems. This subsection investigates
the effect of different smooth parameters $\mu$ for both the STCH-Set scalarized function and the
optimization performance.

**Scalarized Objective Function**    In addition to non-smoothness, the highly non-convex nature of
the $\texttt{min}$ operator will also lead to a poor optimization performance of TCH-Set. As can be found in
Figure 4(a), the $\texttt{min}$ of three convex functions can be highly non-convex, and a gradient-based method
might lead to a bad local optimum. This is not just the case for TCH-Set, but also for SoM (Ding
et al., 2024), which generalizes the $K$-mean clustering algorithm for optimization. It is well-known
that the $K$-mean clustering is NP-Hard. On the other hand, as shown in Figure 4(b), the smooth
version of $\texttt{min}$ we used can lead to a better optimization landscape, especially with a larger $\mu$. When
$\mu$ becomes smaller, the smooth function $\texttt{smin}$ will converge to the original non-convex $\texttt{min}$ function.

**Optimization Performance**    We have conducted a new ablation study for STCH-Set with different
fixed smoothing parameters $\mu = \{10, 5, 1, 0.5, 0.1\}$ and an adaptive schedule from large to small
$\mu$ in Table 8. In this paper, the adaptive schedule we use is $\mu(t) = \exp(-3 \times 10^{-3}t)$, which
gradually reduce from 1 to 0.05 with $t = 10,000$. According to the results, STCH-Set with
adaptive large-to-small $\mu$ can achieve the best performance. This strategy is also related to homotopy
optimization (Dunlavy & O'Leary, 2005; Hazan et al., 2016), which optimizes a gradual sequence of
problems from easy surrogates to the hard original problem.

## D.2 IMPACT OF DIFFERENT PREFERENCES

Table 9: Results of STCH-Set with different preferences on the convex many-objective optimization problem with $K = 1,024$ solutions.

| | $K = 3$ | | $K = 5$ | | $K = 10$ | | $K = 20$ | |
|---|---|---|---|---|---|---|---|---|
| | worst | average | worst | average | worst | average | worst | average |
| Preference from $Dir([1]^m)$ | 4.35e+00 | 4.27e-01 | 2.75e+00 | 2.10e-01 | 2.56e+00 | 1.39e-01 | 2.11e+00 | 1.02e-01 |
| Preference from $Dir([5]^m)$ | 4.22e+00 | 4.09e-01 | 2.22e+00 | 1.98e-01 | 8.43e-01 | 8.28e-02 | 5.66e-01 | 2.61e-02 |
| Preference from $Dir([10]^m)$ | 3.90e+00 | 4.04e-01 | 1.74e+00 | 1.92e-01 | 7.08e-01 | 6.52e-02 | 4.48e-01 | 2.48e-02 |
| Uniform Preference | **1.87e+00** | **3.92e-01** | **9.93e-01** | **1.87e-01** | **4.05e-01** | **5.94e-02** | **2.28e-01** | **2.27e-02** |

The preference $\lambda$ is also a hyperparameter in STCH-Set. In this paper, we mainly focus on the key few-for-many setting, and use the simple uniform preference for all experiments. We believe that the preference $\lambda$ in the (S)TCH-Set can provide more flexibility to the user, especially when they have a specific preference among the objectives. More theoretical analysis can also be investigated with a connection to the weighted clustering (Ackerman et al., 2012).

We have conducted an experiment to investigate the performance of STCH-Set with different preferences sampled from different Dirichlet distributions $Dir([d]^m)$. $Dir([1]^m)$ is also called the uniform Dirichlet distribution, where the preference is diverse and uniformly sampled from the preference simplex. When the $d$ becomes larger, the probability density will concentrate to the middle of the simplex, and therefore the preference for different objectives could be more similar to each other. In the extreme case, it converges to the fixed uniform preference we used in the paper. According to the results in Table 9, if our final goal is still to optimize the unweighted worst objective, we should assign equal preference for all objectives as in the paper. However, the $\lambda$ could be very useful if the decision-maker has a specific preference for different objectives, where the final goal is not to optimize the unweighted worst performance.

We hope our work can inspire more interesting follow-up work that investigates the impact of different preferences, such as preference-driven (S)TCH-Set, adaptive preference adjustment, and inverse preference inference from the optimal solution set.

## D.3 RUNTIME AND SCALABILITY

Table 10: Runtime for different algorithms on the convex multi-objective optimization problem.

| | LS | TCH | STCH | MosT | SoM | TCH-Set | STCH-Set |
|---|---|---|---|---|---|---|---|
| $m = 128$ | 56s | 58s | 59s | 5500s | 58s | 57s | 58s |
| $m = 1024$ | 59s | 62s | 61s | - | 62s | 61s | 63s |

The (S)TCH-Set approach proposed in this work is a pure scalarization methods. Just like other simple scalarization methods, it does not have to separately calculate and update the gradient for each objective with respect to each solution. What (S)TCH-Set has to do is to directly calculate the gradient of the single scalarized value and then update the solutions with simple gradient descent. Informally, if we treat the long vector that concatenates all solutions as a single large solution $x_{\text{all}} = [x1, x2, ..., x_K]$, STCH-Set just define a scalarized value of different objectives with respect to $x_{\text{all}} \in R^{nK}$. It shares the same computational complexity with simple linear scalarization with $m$ objective function on a solution $x \in R^{nK}$, and can scale well to handle a large number of objectives.

In contrast, the gradient manipulation methods like multiple gradient decent algorithm (MGDA) will have to explicitly calculate the gradient for all objectives with respect to all solutions. Therefore, the MosT method that extends MGDA has to separately calculate and manipulate all $mK$ gradients at each iteration, which will scale poorly with the number of objectives.

We report the runtime of different methods for the convex many-objective optimization in Table 10. According to the results, TCH-Set and STCH-Set both have similar runtime with simple linear scalarization, which will not significantly increase with the number of objectives $m$. In contrast, the

MosT method that leverages the MGDA approach will require a significantly longer runtime for $m = 128$, and cannot be conducted on the $m = 1,024$ case with a reasonable run time.

## D.4 Better Performance over SoM

In the experiments, we observe that STCH-set can outperform SoM (Ding et al., 2024) on the average metric in most cases, although SoM is designed to optimize the average metric. The performance advantage of STCH-Set over SoM can be analyzed from the following two perspectives:

**Viewpoint of Optimization** In the ideal case, SoM should achieve the best performance if the final goal is to optimize the average metric. However, similar to the discussion in Appendix D.1, due to the non-smooth `min` operator, SoM also suffers from a slow convergence rate and has a highly non-convex optimization landscape, which might lead to a relatively worse final performance.

**Viewpoint of Clustering** According to the SoM paper (Ding et al., 2024), SoM can be treated as a generalization of the classic $K$-means clustering problem with Lloyd's algorithm by alternately updating the clusters (assign one of $K$ solutions to each objective) and their centroids (update the solution for each group). However, the $K$-means clustering is well known to be NP-hard, and hence SoM is not guaranteed to assign the most suitable solution to each objective. From the experimental results, we can find that SoM will be outperformed by STCH-Set in most cases with a larger number of $K$, which might caused by a wrong solution-objective matching during the optimization process (which could be harder for a larger $K$). To some degree, our proposed STCH-Set can also be treated as a smooth clustering method (in contrast to the hard $K$-means clustering). We leave this interesting research direction to future work.

## D.5 Setting the Number of Solutions

The number of solutions $K$ is a hyperparameter in the proposed (S)TCH-Set scalarization, which can be properly set by:

**Following the Requirement from Specific Application** In many real-world few-for-many applications, such as 1) training a few models for different tasks and 2) producing a few different versions of advertisements to serve diverse audiences, the number of solutions will be naturally required and limited by the (computational and financial) budget.

**Conducting Hyperparameter Search** As shown in the experimental results, there is a clear trade-off between the number of solutions and the performance of (S)TCH-Set. More solutions can lead to better performance, but they also require a larger (computational and financial) budget. Therefore, it is reasonable to solve the (S)TCH-Set optimization problem with different $K$ and choose the most suitable number of solutions that satisfy the decision-maker's preference.

More advanced methods, such as those for selecting the proper number of clusters in clustering, might also be generalized to handle this similar issue for (S)TCH-Set. We hope this work can inspire more follow-up work on efficient $K$ selection for few-for-many optimization.

## D.6 Discussion with Dimensionality Reduction

Dimensionality reduction (Deb & Saxena, 2005; Brockhoff & Zitzler, 2006; Singh et al., 2011) is a widely used technique to deal with many-objective optimization problems with potential redundant objectives. By summarizing all objectives by a few representative objectives, these methods can reformulate the originally challenging problem into a simpler problem with much fewer objectives. For example, in Figure 2 of the main paper, we show the result of our proposed method on a synthetic optimization problem with 100 objectives. To clearly demonstrate the behavior of our proposed method, we intentionally construct the problem such that it has 5 groups of very similar 20 objectives. Therefore, from the viewpoint of dimensionality reduction, these 20 objectives can be represented by a single objective. In this way, the original 100-objective problem can be summarized by a 5-objective problem.

On the other hand, the solution set optimization we investigated in this paper also has it own advantages for many-objective optimization.

**Requirement from Real-World Application:** The requirement of finding a few solutions to tackle many optimization objectives will naturally arise in many real-word applications such as building a small set of machine learning models (e.g., mixture of experts) to handle many different data or tasks, training a few agents to handle a large number of clients in federated learning, and producing a few different versions of advertisements to serve a large group of diverse audiences. For these cases, it is more natural to tackle the problem from the viewpoint of solution set optimization.

**No Redundant Objectives:** In many real-world problems, the objectives we care about are not redundant or not highly correlated with each other. By keeping all objectives, the solution set optimization method can more flexibly tackle the trade-offs among all different objectives. For example, when all objectives are conflicting, our proposed (S)TCH-Set will actually find a Pareto solution with the optimal (S)TCH scalarization value for each group of objectives while the groups are adaptively assigned during the optimization process. If the best solution-objective group assignment is known, it is analogous to running the traditional (S)TCH scalarization to find a Pareto solution for each group of objectives. The flexibility of the few-solutions-for-many-objectives approach over the few-solutions-for-few-summarized-objectives is also an interesting research topic in future work.

**Bridging Different Approaches:** To some degree, our proposed (S)TCH-Set method can be treated as a generalized (smooth) clustering method that assigns each solution to tackle different groups (clusters) of solutions. If we only keep one representative objective for each group, it could become a dimensionality reduction approach at the end. In addition, if we treat the proposed (S)TCH-Set scalarization function as an indicator, it is actually an indicator-based method for many-objective optimization. We hope this paper can inspire more interesting follow-up works on bridging these different but related approaches.

### D.7    MORE DISCUSSION ON THE MOTIVATION AND PRACTICAL IMPACT

Our proposed (S)TCH-Set method is motivated by the need of finding a few complementary solutions to tackle many different optimization objectives that naturally arise in many real-world applications. Some typical applications include building a small set of machine learning models (e.g., mixture of experts) to handle many different data or tasks, training a few agents to handle a large number of clients in federated learning, and producing a few different versions of advertisements to serve a large group of diverse audiences. A single solution with a balanced trade-off is not enough to properly address these applications. Very recently, this problem setting has also been investigated in two concurrent works (Ding et al., 2024; Li et al., 2024) by researchers with backgrounds in machine learning, traditional optimization, and federated learning.

Due to the requirement on finding a set of complementary solutions, traditional multi-objective optimization methods cannot be directly used to solve these problems. In contrast, our proposed (S)TCH-Set method can efficiently handle this problem setting. The experimental results also show our proposed (S)TCH-Set can outperform the methods proposed in (Ding et al., 2024; Li et al., 2024; Standley et al., 2020; Fifty et al., 2021) on different application problems. Some application problems considered in this work, such as multi-task grouping (Standley et al., 2020; Fifty et al., 2021), are already impactful in practice. We believe our proposed method will have a significant practical impact on solving a broad range of similar application problems that require a set of complementary solutions.

### D.8    RELATION WITH TRADITIONAL METHODS

Rooted in multi-objective optimization, our proposed (S)TCH-Set method is a complement rather than a replacement for the traditional multi-objective optimization method:

- The proposed (S)TCH-Set scalarization is a natural extension of traditional multi-objective optimization method. If we set the number of solutions to $K = 1$, it will reduce to the classic TCH-Set scalarization as well as the smooth TCH-Set scalarization Lin et al. (2024) that aims to find a single Pareto solution with a balanced trade-off.

- If we treat the (S)TCH-Set scalarization function as an indicator, our proposed method is indeed a standard indicator-based method for many-objective optimization. Here, we want to find a (small) set of solutions to optimize the newly proposed (S)TCH-Set scalarization value rather than the values of traditional indicators (e.g., Hypervolume).

## E   NOTATION TABLE

Table 11: Notation Table

| Notation | Definition |
|---|---|
| $m$ | number of objectives |
| $n$ | dimension of solutions |
| $K$ | number of solutions |
| $\boldsymbol{x}$ | solution |
| $\boldsymbol{X}$ | solution set |
| $\boldsymbol{f}(\boldsymbol{x})$ | objective vector |
| $f_i(\boldsymbol{x})$ | i-th objective value |
| $\nabla f_i(\boldsymbol{x})$ | gradient of the i-th objective |
| $\mathcal{X}$ | decision space |
| $\lambda$ | preference |
| $\boldsymbol{\Delta}^{m-1}$ | preference simplex |

