# OpenReview forum: "Few for Many: Tchebycheff Set Scalarization for Many-Objective Optimization"
_ICLR.cc/2025/Conference — ICLR 2025 Poster_

### Official Review · Reviewer_u9MJ · 2024-11-01

**Soundness:** 2
**Presentation:** 3
**Contribution:** 3
**Rating:** 6
**Confidence:** 2

**Summary:**

Authors propose a novel Tchebycheff set scalarization method to find a few representative solution to cover a large number of objectives
 in a collaborative and complementary manner so that each objective can be well addressed by at least one solution in the small solution set.

**Strengths:**

* This paper is well-written and easy to follow.

* The experiments are comprehensive and include necessary analyses, and the results are convincing.

* The necessary proofs are included.

**Weaknesses:**

* This novelty is somewhat limited. It seems the line 74 “a novel Tchebycheff set (TCH-Set) scalarization approach”  has been proposed in [1] and the line 75 “a smooth Tchebycheff set (STCH-Set) scalarization approach” has been proposed in [2].

* The motivation is not quite clear. Why should you find a few representative solutions to cover a large number of objectives?

* The work is not tied to particular applications. Can you briefly provide some scenarios where your method is applicable?

**Questions:**

* The experiments are conducted with different K, how to choose the K of your algorithm?

* Since the preference $\lambda$ can greatly impacts the final result, could you please provide with some hints for choosing  $\lambda$?

Reference：
1.  Eng Ung Choo and DR Atkins. Proper efficiency in nonconvex multicriteria programming. Mathematics of Operations Research, 8(3):467–470, 1983.
2.  Xi Lin, Xiaoyuan Zhang, Zhiyuan Yang, Fei Liu, Zhenkun Wang, and Qingfu Zhang. Smooth tchebycheff scalarization for multi-objective optimization. arXiv preprint arXiv:2402.19078, 2024.

---

> ### Author Response · Authors · 2024-11-24
> **Response to Reviewer u9MJ [1/3]**
>
> Thank you very much for your time and effort in reviewing our work. Our responses to your concerns are as follows.
>
> > **W1. Relation to Previous Work:** This novelty is somewhat limited. It seems the line 74 “a novel Tchebycheff set (TCH-Set) scalarization approach” has been proposed in [1] and the line 75 “a smooth Tchebycheff set (STCH-Set) scalarization approach” has been proposed in [2].
>
> > [1] Eng Ung Choo and DR Atkins. Proper efficiency in nonconvex multicriteria programming. Mathematics of Operations Research, 8(3):467–470, 1983.
>
> > [2] Xi Lin, Xiaoyuan Zhang, Zhiyuan Yang, Fei Liu, Zhenkun Wang, and Qingfu Zhang. Smooth tchebycheff scalarization for multi-objective optimization. arXiv preprint arXiv:2402.19078, 2024.
>
>
>
> We believe there is a misunderstanding here regarding our contribution. In this work, our proposed *(S)TCH-**Set*** approaches significantly extend the traditional *(S)TCH* approaches [1,2] to tackle the novel and important few-for-many problem setting for many-objective optimization.
>
> - The scalarization methods proposed in [1,2] are to find a single Pareto solution with a specific trade-off among all objectives. The work [1] proposed the classic Tchebycheff (TCH) scalarization, while the work [2] proposed a smooth version (STCH) for efficient gradient-based optimization. Both of them cannot tackle the few-solutions-for-many-objectives problem setting considered in this work.
>
> - In this work, we propose the (S)TCH-Set scalarization method to find a small set of solutions to cover a large number of objectives in a collaborative and complementary way such that each optimization objective can be well addressed by at least one solution. This problem setting is important for many real-world applications (see discussions below).
>
> - The (S)TCH-Set scalarization is an important and non-trivial extension of the classic TCH-Set scalarization, and it will naturally reduce to TCH-Set scalarization with the number of solutions $K = 1$. We have also provided a detailed theoretical analyses to show that our proposed (S)TCH-Set approach enjoy good theoretical properties for multi-objective optimization.
>
> - Experimental results show our proposed (S)TCH-Set can significantly outperform the classic (S)TCH as well as two very recently proposed methods on multi-objective optimization problems with many objectives.
>
> Therefore, we believe our proposed (S)TCH-Set scalarization method is a novel contribution to many-objective optimization.
>
> [1] Eng Ung Choo and DR Atkins. Proper efficiency in nonconvex multicriteria programming. Mathematics of Operations Research, 8(3):467–470, 1983.
>
> [2] Xi Lin, Xiaoyuan Zhang, Zhiyuan Yang, Fei Liu, Zhenkun Wang, and Qingfu Zhang. Smooth tchebycheff scalarization for multi-objective optimization. arXiv preprint arXiv:2402.19078, 2024.

---

> > ### Author Response · Authors · 2024-11-24
> > **Response to Reviewer u9MJ [2/3]**
> >
> > > **W2. Motivation:** The motivation is not quite clear. Why should you find a few representative solutions to cover a large number of objectives?
> >
> > > **W3. Application:** The work is not tied to particular applications. Can you briefly provide some scenarios where your method is applicable?
> >
> > Thank you for raising this concern. Our proposed (S)TCH-Set method is motivated by the need of finding a few complementary solutions to tackle many different optimization objectives that naturally arise in many real-world applications. Some typical applications include building a small set of machine learning models (e.g., mixture of experts) to handle many different data or tasks, training a few agents to handle a large number of clients in federated learning, and producing a few different versions of advertisements to serve a large group of diverse audiences. A single solution with a balanced trade-off is not enough to properly address these applications. Very recently, this problem setting has also been investigated in two concurrent works [3,4] by researchers with backgrounds in machine learning, traditional optimization, and federated learning. Therefore, we believe the problem setting considered in this work is valid and important for many real-world applications.
> >
> > Due to the requirement on finding a set of complementary solutions, traditional multi-objective optimization methods cannot be directly used to solve these problems. In contrast, our proposed (S)TCH-Set method can efficiently handle this problem setting. The experimental results also show our proposed (S)TCH-Set can outperform the methods proposed in [3,4,5,6] on different application problems. Some application problems considered in this work, such as multi-task grouping [5,6], are already impactful in practice. We believe our proposed method will have a significant practical impact on solving a broad range of similar application problems that require a set of complementary solutions.
> >
> > The above discussion has been added in Appendix D.7 of the revised paper.
> >
> > [3] Lisang Ding, Ziang Chen, Xinshang Wang, Wotao Yin. Efficient Algorithms for Sum-of-Minimum Optimization. ICML 2024.
> >
> > [4] Ziyue Li, Tian Li, Virginia Smith, Jeff Bilmes, Tianyi Zhou. Many-Objective Multi-Solution Transport. arXiv:2403.04099.
> >
> > [5] Trevor Standley, Amir R. Zamir, Dawn Chen, Leonidas Guibas, Jitendra Malik, Silvio Savarese. Which Tasks Should Be Learned Together in Multi-task Learning? ICML 2020.
> >
> > [6] Christopher Fifty, Ehsan Amid, Zhe Zhao, Tianhe Yu, Rohan Anil, Chelsea Finn. Efficiently Identifying Task Groupings for Multi-Task Learning. NeurIPS 2021.
> >
> > > **Q1. Choosing $K$:** The experiments are conducted with different K, how to choose the K of your algorithm?
> >
> > The number of solutions $K$ is a hyperparameter in the proposed (S)TCH-Set scalarization, which can be properly set by:
> >
> > **Following the Requirement from Specific Application.** In many real-world few-for-many applications, such as 1) training a few models for different tasks and 2) producing a few different versions of advertisements to serve diverse audiences, the number of solutions will be naturally required and limited by the (computational and financial) budget.
> >
> > **Conducting Hyperparameter Search.** As shown in the experimental results, there is a clear trade-off between the number of solutions and the performance of (S)TCH-Set. More solutions can lead to better performance, but they also require a larger (computational and financial) budget. Therefore, it is reasonable to solve the (S)TCH-Set optimization problem with different $K$ and choose the most suitable number of solutions that satisfy the decision-maker's preference.
> >
> > More advanced methods, such as those for selecting the proper number of clusters in clustering, might also be generalized to handle this similar issue for (S)TCH-Set. We hope this work can inspire more follow-up work on efficient $K$ selection for few-for-many optimization.
> >
> > The above discussion can be found in Appendix D.5 of the revised paper.

---

> > > ### Author Response · Authors · 2024-11-24
> > > **Response to Reviewer u9MJ [3/3]**
> > >
> > > > **Q2. Preference:** Since the preference can greatly impacts the final result, could you please provide with some hints for choosing the preference?
> > >
> > > The preference $\lambda$ is a also hyperparameter in (S)TCH-Set. In this paper, we mainly focus on the key few-for-many setting, and use the simple uniform preference for all experiments. We believe that the preference term in the (S)TCH-Set can provide more flexibility to the user, especially when they have a specific preference among the objectives. More theoretical analysis can also be investigated with a connection to the weighted clustering [7].
> > >
> > > |                               |   K = 3  |          |   K = 5  |          |  K = 10  |          |   K=20   |          |
> > > |:-----------------------------:|:--------:|:--------:|:--------:|:--------:|:--------:|:--------:|:--------:|:--------:|
> > > |                               |   worst  |  average |   worst  |  average |   worst  |  average |   worst  |  average |
> > > | Preference from $Dir([1]^m)$  | 4.35e+00 | 4.27e-01 | 2.75e+00 | 2.10e-01 | 2.56e+00 | 1.39e-01 | 2.11e+00 | 1.02e-01 |
> > > | Preference from $Dir([5]^m)$  | 4.22e+00 | 4.09e-01 | 2.22e+00 | 1.98e-01 | 8.43e-01 | 8.28e-02 | 5.66e-01 | 2.61e-02 |
> > > | Preference from $Dir([10]^m)$ | 3.90e+00 | 4.04e-01 | 1.74e+00 | 1.92e-01 | 7.08e-01 | 6.52e-02 | 4.48e-01 | 2.48e-02 |
> > > | Uniform Preference            | 1.87e+00 | 3.92e-01 | 9.93e-01 | 1.87e-01 | 4.05e-01 | 5.94e-02 | 2.28e-01 | 2.27e-02 |
> > >
> > > We have conducted an experiment to investigate the performance of STCH-Set with different preferences sampled from different Dirichlet distributions $Dir([d]^m)$. $Dir([1]^m)$ is also called the uniform Dirichlet distribution, where the preference is diverse and uniformly sampled from the preference simplex. When the $d$ becomes larger, the probability density will concentrate to the middle of the simplex, and therefore the preference for different objectives could be more similar to each other. In the extreme case, it converges to the fixed uniform preference we used in the paper. According to the results in the table above, if our final goal is still to optimize the unweighted worst objective, we should assign equal preference for all objectives as in the paper. However, the $\lambda$ could be very useful if the decision-maker has a specific preference for different objectives, where the final goal is not to optimize the unweighted worst performance.
> > >
> > > We hope this work can inspire more interesting follow-up work that investigates the impact of different preferences, such as preference-driven(S)TCH-Set, adaptive preference adjustment, and inverse preference inference from the optimal solution set.
> > >
> > > [7] Margareta Ackerman, Shai Ben-David, Simina Brânzei, David Loker. Weighted Clustering. AAAI 2012.
> > >
> > > The above discussion can be found in Appendix D.2 of the revised paper.

---

> ### Comment · Reviewer_u9MJ · 2024-11-26
> **Thank you for your response**
>
> Thank you for the author's response. I consider this work to be rigorous and thorough. Most of my concerns have been resolved, hence I have increased my score.
>
> I am not very familiar with the field of multi-objective optimization, but I hope to see the author's research have more potential applications in practical scenarios.

---

> ### Author Response · Authors · 2024-11-26
>
> We are very glad to know most of your concerns have been resolved, and you consider this work to be rigorous and thorough.
>
> We are currently trying to apply TCH-Set to tackle more real-world applications. Given the workload, new experimental results would not be available during the rebuttal period. We will try our best to include more experimental evaluation on new tasks in the revised paper, and explore the potential of TCH-Set to tackle different real-world applications in future work.
>
> Thank you again for your time and effort in reviewing our work.

---

### Official Review · Reviewer_Uihz · 2024-11-01

**Soundness:** 4
**Presentation:** 3
**Contribution:** 4
**Rating:** 8
**Confidence:** 4

**Summary:**

Different from general multi-objective optimization aiming to find a dense Pareto set, this paper focuses on identifying a few solutions that cover a large number of objectives in a complementary manner for many-objective optimization. To achieve this, a smooth Tchebycheff set scalarization approach is developed with good theoretical guarantees.

**Strengths:**

1. This paper introduces a novel and practical prospective in tackling for many-objective optimization.
2. This paper develops a smooth Tchebycheff set scalarization approach under the few-for-many prospective.
3. The theoretical analysis is thorough and well-supported.

**Weaknesses:**

Some descriptions lack clarity, and minor writing errors are present. Please refer to the questions below for the details.

**Questions:**

1. After the Tchebycheff set scalarization, how is the scalarized problem solved, given that a set of solutions needs to be optimized?
2. Did the methods compared in the experiments also propose the few-for-many prospective for many-objective optimization?
3. What optimization methods are used under the scalarization methods for the convex multi-objective optimization in Section 4.1?
4. A brief analysis of why STCH-Set performs slightly worse than SoM in two cases in Table 4 would be beneficial.
5. It should be clarified why the part results of MosT are “-” in Table 6.
6. All mathematical symbols (e.g., $K$ and $m$) in the tables should use the correct mathematical formatting.
7. In line 227, a space is missing after the period.
8. In line 1473, there is an incorrect line break, and “term0” appears to be a typo.
9. In line 1487, a space is missing before “(S)TCH”.
10. An open question is whether the proposed approach is especially suited for many-objective optimization with non-conflicting objectives. How would it perform when all objectives are conflicting?

---

> ### Author Response · Authors · 2024-11-24
> **Response to Reviewer Uihz [1/2]**
>
> Thank you very much for your time and effort in reviewing our work. Our responses to your concerns are as follows.
>
> > **Q1. Optimization for (S)TCH-Set:** After the Tchebycheff set scalarization, how is the scalarized problem solved, given that a set of solutions needs to be optimized?
>
> In short, all solutions are simply optimized together by a gradient-based method.
>
> - If we treat the (S)TCH-Set scalarization function as an indicator, our proposed method is indeed a standard indicator-based method for many-objective optimization.
>
> - To minimize the scalarized (S)TCH-Set value, we treat the set of solutions as a whole (i.e. a solution matrix) and optimize them together using a gradient-based optimization method as shown in Algorithm 1 in the paper.
>
> We have revised the related description for algorithm 1 to make this point more clear.
>
> > **Q2. Few-for-Many Prospective:** Did the methods compared in the experiments also propose the few-for-many prospective for many-objective optimization?
>
> Yes, some methods also propose the few-for-many problem setting, but not necessarily consider many-objective optimization. In the experiments, we compare our proposed (S)TCH-Set with traditional multi-objective optimization algorithms as well as two recently proposed methods that aim to find a small set of solutions to tackle many optimization objectives. Among them:
>
> - The traditional multi-objective optimization methods, such as linear scalarization and (smooth) Tchebycheff scalarization, focus on finding solutions with specific trade-offs and do not consider the few-for-many problem setting.
>
> - The SoM method generalizes the classic k-means clustering method to handle a specific Sum-of-Minimization (SoM) problem $\frac{1}{m} \sum\_{i=1}^m \min \{f_i(x^{(1)}), f_i(x^{(2)}), \ldots, f_i(x^{(K)})\}$ with a few solutions $\\{x^{(k)}\\}\_{k=1}^K$ and many optimization functions $\\{f_i(\cdot)\\}\_{i=1}^m$. However, it does not take multi-objective optimization into consideration.
>
> - The Many-objective multi-solution Transport (MosT) method leverages bi-level optimization, optimal transport, and MGDA to find a small set of diverse solutions on different representative regions of the
> Pareto front. However, unlike SoM and our (S)TCH-Set method, it does not have an indicator to explicitly take each objective value into consideration.
>
> In this work, we propose a straightforward and efficient set scalarization approach to explicitly optimize all objectives by a small set of solutions for many-objective optimization.
>
> >**Q3. Optimization Methods:** What optimization methods are used under the scalarization methods for the convex multi-objective optimization in Section 4.1?
>
> We use a simple SGD optimizer with learning rate $0.01$ to all methods for the convex multi-objective optimization problem.
>
> >**Q4. STCH-Set and SoM:** A brief analysis of why STCH-Set performs slightly worse than SoM in two cases in Table 4 would be beneficial.
>
> Thank you for this valuable suggestion.
>
> - Table 4 reports the results for a real-world deep multi-task grouping problem with $9$ tasks (objectives) and $\{2, 3, 4\}$ models (solutions). Our proposed STCH-Set method can outperform SoM in all cases on the worst-case performance, but is slightly worse than SoM in two cases on the average performance.
>
> - SoM is designed to explicitly optimize the average performance by its definition. Therefore, in the ideal case, SoM should always achieve the best average performance if it has been properly optimized. However, due to the non-smoothness of the SoM objective and NP-hard nature of the clustering algorithm, SoM is outperformed by STCH-Set on the average performance in most cases for the convex optimization and mixed (non)linear regression problem with a large number of objectives ($128$ to $1024$). A brief discussion can be found in Appendix D.4.
>
> - For the deep multi-task grouping problem, we only have a relatively small number of objectives ($9$). In this case, it seems that SoM can properly assign a suitable solution to each objective much more easily than those with many more objectives, which leads to a good average performance. By definition, our proposed (S)TCH-Set is to minimize the worst-case performance among all objectives. Therefore, it is inevitable that STCH-Set can not achieve the best average performance at the same time for a non-trivial problem.
>
> The above discussion has been added for analyzing the results of multi-task grouping in the revised paper.

---

> > ### Author Response · Authors · 2024-11-24
> > **Response to Reviewer Uihz [2/2]**
> >
> > > **Q5. Results of MosT:** It should be clarified why the part results of MosT are “-” in Table 6.
> >
> > - The MosT method relies on MGDA to find multiple solutions to cover different representative regions of the Pareto front, which will scale poorly with the number of objectives. As reported in Table 11 in Appendix D3, the MosT method will already require a significantly longer runtime for the problem with $128$ objectives.
> >
> > - Since the MosT method cannot tackle the problems with $1024$ objectives in a reasonable time, we did not include these results in Table 6 and marked it as "-".
> >
> > The above clarification has been added in the revised paper for Table 6.
> >
> > > **Q6. Symbol in Table:** All mathematical symbols (e.g., $K$ and $m$) in the tables should use the correct mathematical formatting.
> >
> > Thank you for this valuable suggestion. We have now used the correct mathematical formatting for all the mathematical symbols in the tables.
> >
> > > **Q7/8/9. Typos:** In line 227, a space is missing after the period.
> >
> > > In line 1473, there is an incorrect line break, and “term0” appears to be a typo.
> >
> > > In line 1487, a space is missing before “(S)TCH”.
> > \end{tcolorbox}
> >
> > Thank you very much for pointing them out. Now all the typos have been fixed in the revised paper.
> >
> > > **Q10. Conflicting Objectives:** An open question is whether the proposed approach is especially suited for many-objective optimization with non-conflicting objectives. How would it perform when all objectives are conflicting?
> >
> >
> > Thank you for this valuable and important open question.
> >
> > - By definition, no matter whether all objectives are conflicting or not, our proposed method will optimize the worst-case performance among all objectives (e.g., the (S)TCH-Set scalarization value). It is analogous to the other indicator-based methods that will always optimize their corresponding indicator (e.g., hypervolume).
> >
> > - If the objectives are less conflicting with each other, (S)TCH-Set can find a set of solutions that achieve (nearly) optimal performance for all objectives. In this case, it will be closely related to the dimensionality reduction method for many-objective optimization with redundant objectives. Please also see our response to reviewer CNbj for a detailed discussion on dimensionality reduction.
> >
> > - When all objectives are conflicting, (S)TCH-Set will actually find a Pareto solution with the optimal (S)TCH scalarization value for each group of objectives while the groups are adaptively assigned during the optimization process. If the best solution-objective group assignment is known, it is analogous to running the traditional (S)TCH scalarization to find a Pareto solution for each group of objectives.
> >
> > The above discussion has been added in Appendix D.6 of the revised paper.

---

> > > ### Comment · Reviewer_Uihz · 2024-11-26
> > >
> > > I appreciate the authors' detailed responses and their efforts to enhance the quality of this work. After reviewing all the comments from the other reviewers and the authors' responses, I find that all of my concerns have been fully addressed. Given the new 'few-for-many' perspective and the solid theoretical contributions, I keep my positive score and fully support the acceptance of this paper.

---

> > > > ### Author Response · Authors · 2024-11-26
> > > >
> > > > We are very glad to know that all of your concerns have been fully addressed. We really appreciate your positive score and support for our work.
> > > >
> > > > Thank you again for your time and effort in reviewing our work.

---

### Official Review · Reviewer_m2ed · 2024-11-03

**Soundness:** 3
**Presentation:** 3
**Contribution:** 2
**Rating:** 6
**Confidence:** 4

**Summary:**

This paper proposes a novel scalarization method called TCH-Set to produce a limited number of solutions that handle many objectives. A relaxed version is also proposed to tackle the non-smooth situations. The theoretical properties of the proposed method are investigated, and an empirical study is conducted to verify its effectiveness.

**Strengths:**

1. This paper is well-written and easy to follow.
2. The proposed method is well-motivated.
3. The technical details are clearly presented, and the theoretical analyses seem correct.
4. The literature review is comprehensive.

**Weaknesses:**

My major concern is about the problem setting. The proposed method aims to find a few solutions that collaboratively optimize many objectives. This problem setting is quite different from the common setting in multi- or many-objective optimization, which aims to find some solutions with diverse trade-offs. I agree with the authors that one of the main obstacles in many-objective optimization is that, with the increase in the number of objectives, the number of solutions has to increase exponentially to cover the whole PF. The idea proposed in this paper is interesting; however, I do not think it really solves this problem. When the number of objectives is very high, it only searches for extreme points, that is, solutions with very low values for some objectives but significant sacrifices in others. By putting together these multiple extreme points, it appears as if multiple objective functions can be *"simultaneously"* optimized. Actually, although many MOAs output a set of solutions, after multi-objective decision-making, typically only one solution is finally selected, and, in practice, solutions with a more balanced tradeoff are often preferred. Therefore, while the TCH-Set offers a novel idea for many-objective optimization, I think its practical impact and significance are limited.

**Questions:**

Please refer to "Weaknesses". Additionally, I suggest the authors summarize a notation table in the appendix.

---

> ### Author Response · Authors · 2024-11-24
> **Response to Reviewer m2ed [1/2]**
>
> > **W1. Problem Setting:** My major concern is about the problem setting. The proposed method aims to find a few solutions that collaboratively optimize many objectives. This problem setting is quite different from the common setting in multi- or many-objective optimization, which aims to find some solutions with diverse trade-offs. I agree with the authors that one of the main obstacles in many-objective optimization is that, with the increase in the number of objectives, the number of solutions has to increase exponentially to cover the whole PF. The idea proposed in this paper is interesting; however, I do not think it really solves this problem. When the number of objectives is very high, it only searches for extreme points, that is, solutions with very low values for some objectives but significant sacrifices in others. By putting together these multiple extreme points, it appears as if multiple objective functions can be "simultaneously" optimized. Actually, although many MOAs output a set of solutions, after multi-objective decision-making, typically only one solution is finally selected, and, in practice, solutions with a more balanced tradeoff are often preferred. Therefore, while the TCH-Set offers a novel idea for many-objective optimization, I think its practical impact and significance are limited.
>
> Thank you very much for raising this valuable concern. First of all, we want to be very clear that our proposed method is not to replace the traditional methods that find a single or a set of solutions with balanced trade-offs. In contrast, it provides a new approach to tackle a type of application problem that was originally not considered by the traditional many-objective optimization method. Therefore, our proposed method is a complement rather than a replacement for the traditional methods for many-objective optimization.
>
> We provide the following discussion on 1) the problem setting and practical impact of our method and 2) its relation with traditional methods to address your concerns.
>
> **Problem Setting and Practical Impact.** Our proposed (S)TCH-Set method is motivated by the need of finding a few complementary solutions to tackle many different optimization objectives that naturally arise in many real-world applications. Some typical applications include building a small set of machine learning models (e.g., mixture of experts) to handle many different data or tasks, training a few agents to handle a large number of clients in federated learning, and producing a few different versions of advertisements to serve a large group of diverse audiences. A single solution with a balanced trade-off is not enough to properly address these applications. Very recently, this problem setting has also been investigated in two concurrent works [1,2] by researchers with backgrounds in machine learning, traditional optimization, and federated learning. Therefore, we believe the problem setting considered in this work is valid and important for many real-world applications.
>
> Due to the requirement on finding a set of complementary solutions, traditional multi-objective optimization methods cannot be directly used to solve these problems. In contrast, our proposed (S)TCH-Set method can efficiently handle this problem setting. The experimental results also show our proposed (S)TCH-Set can outperform the methods proposed in [1,2,3,4] on different application problems. Some application problems considered in this work, such as multi-task grouping [3,4], are already impactful in practice. We believe our proposed method will have a significant practical impact on solving a broad range of similar application problems that require a set of complementary solutions.
>
> In addition, our proposed (S)TCH-Set method provide a novel viewpoint of multi-objective optimization to tackle the few-for-many problem setting. We hope it can further bridge these fields, attract practitioners' attention to multi-objective optimization, and also inspire more interesting follow-up work on multi-objective optimization to handle novel application problems.
>
> [1] Lisang Ding, Ziang Chen, Xinshang Wang, Wotao Yin. Efficient Algorithms for Sum-of-Minimum Optimization. ICML 2024.
>
> [2] Ziyue Li, Tian Li, Virginia Smith, Jeff Bilmes, Tianyi Zhou. Many-Objective Multi-Solution Transport. arXiv:2403.04099.
>
> [3] Trevor Standley, Amir R. Zamir, Dawn Chen, Leonidas Guibas, Jitendra Malik, Silvio Savarese. Which Tasks Should Be Learned Together in Multi-task Learning? ICML 2020.
>
> [4] Christopher Fifty, Ehsan Amid, Zhe Zhao, Tianhe Yu, Rohan Anil, Chelsea Finn. Efficiently Identifying Task Groupings for Multi-Task Learning. NeurIPS 2021.

---

> > ### Author Response · Authors · 2024-11-24
> > **Response to Reviewer m2ed [2/2]**
> >
> > **Relation with Traditional Methods.**
> >
> > Rooted in multi-objective optimization, our proposed (S)TCH-Set method is a complement rather than a replacement for the traditional multi-objective optimization method.
> >
> > - The proposed (S)TCH-Set scalarization is a natural extension of traditional multi-objective optimization method. If we set the number of solutions to $K = 1$, it will reduce to the classic TCH-Set scalarization as well as the smooth TCH-Set scalarization [5] that aims to find a single Pareto solution with a balanced trade-off.
> >
> > - If we treat the (S)TCH-Set scalarization function as an indicator, our proposed method is indeed a standard indicator-based method for many-objective optimization. Here, we want to find a (small) set of solutions to optimize the newly proposed (S)TCH-Set scalarization value rather than the values of traditional indicators (e.g., Hypervolume). We have also provided the Pareto optimality/stationary guarantee of our proposed method in this paper.
> >
> > - In addition, the proposed method is closely related to the widely used dimensionality reduction method for many-objective optimization. Please refer to our response to reviewer CNbj for a detailed discussion.
> >
> > In the end, we want to sincerely emphasize that our proposed (S)TCH-Set scalarization method is a complement to the classic multi-objective optimization method, and it is by no mean to "really solve the many-objective optimization problem". We hope it can extend multi-objective optimization method to tackle the important few-for-many problem setting that arise in many real-world applications, and also attract the attention from practitioners/researchers working on these fields to multi-objective optimization.
> >
> > [5] Xi Lin, Xiaoyuan Zhang, Zhiyuan Yang, Fei Liu, Zhenkun Wang, and Qingfu Zhang. Smooth Tchebycheff Scalarization for Multi-Objective Optimization. ICML 2024.
> >
> > The above discussion on the motivation and practical impact, as well as the relation with traditional methods, have been added in Appendix D.7 and D.8 in the revised paper, respectively. We have also revised the conclusion to clearly emphasize our proposed method is a complement rather than a replacement for the traditional multi-objective optimization method.
> >
> > > **Q1. Notation Table:** I suggest the authors summarize a notation table in the appendix.
> >
> >
> > Thank you for this valuable suggestion. We have now added a notation table in Appendix E.

---

> ### Comment · Reviewer_m2ed · 2024-11-26
>
> I greatly appreciate your effort in improving the manuscript and providing detailed responses. With your help, I have a clearer understanding of your work. Now I think this paper matches the standard of ICLR, and I support the acceptance of this paper.
>
> However, I still have some minor suggestions for further improvements or future work. This paper does not seem to present adequate evaluations in downstream tasks. You mentioned some potential applications of TCH-Set, such as MoE. I also think this is a very promising application scenario. Perhaps many audiences would also be curious to know if TCH-Set can demonstrate the same superior performance in these real-world tasks.

---

> > ### Author Response · Authors · 2024-11-26
> >
> > We are very glad to know our responses are useful to you and you support the acceptance of this work.
> >
> > Thank you very much for your valuable suggestion, and we totally agree with you that showing the (superior) performance of TCH-Set on more real-world applications and downstream tasks can further improve the practical impact of this paper. In this work, we mainly followed the setting in [1], and reported the evaluation results on convex optimization, noisy (non)linear mixed regression, and deep multi-task grouping. We are currently trying to apply TCH-Set to tackle more real-world applications. Given the workload, new experimental results would not be available during the rebuttal period. We will try our best to include more experimental evaluation on new tasks in the revised paper, and explore the potential of TCH-Set to tackle different real-world applications in future work.
> >
> > [1] Lisang Ding, Ziang Chen, Xinshang Wang, Wotao Yin. Efficient Algorithms for Sum-of-Minimum Optimization.ICML 2024.88
> >
> > In addition, we noticed that your current score is keeping 3 for this paper. We would like to know if there is any remaining concern we can further address during the rebuttal period.

---

> > > ### Comment · Reviewer_m2ed · 2024-11-26
> > >
> > > I am sorry I forgot to change the score. It has now been updated.

---

> > > > ### Author Response · Authors · 2024-11-26
> > > >
> > > > Thank you very much. We will continually improve the quality of this work.
> > > >
> > > > Thank you again for your time and effort in reviewing our work.

---

### Official Review · Reviewer_CNbj · 2024-11-04

**Soundness:** 4
**Presentation:** 4
**Contribution:** 4
**Rating:** 8
**Confidence:** 3

**Summary:**

The authors propose a scheme for many-objective problems with large number of objectives (e.g > 100) where the idea is to find a few representative solutions that address them.
To make this scheme possible they use their newly proposed Tchebycheff Set scalarization function that is an extension of the original. Now instead of trying to find a point that minimizes the scalarized problem, is trying to find multiple points that minimize it, allowing it to be bad for some of the m objectives, but it should at least minimize some subset of it.
The original Tchebycheff function is non differentiable to the max and min operators, which makes it not the best option to work with gradient based optimization. Therefore they extend the smooth Tchebycheff function in the work Lin et.al 2024 to work on set of solutions.

**Strengths:**

Scalarization that is well known to perform well on multi-objective problems as seen on different variants of the evolutionary algorithm MOEA/D. Tchebycheff as the scalarizing function has also been shown to cover the non-convex solutions in the Pareto Front. The regular approach only aims to have a solution per direction, by modifying the function to instead search for a set of points that while being good for the other objectives it should be the best for a subset of all objectives.

The text introduces well non multi-objective practitioners to the setting, the difficulty present in many-objective problems, and previous work and results on scalarization with the simple weighted sum and the Tchebycheff function.
Appendices further explore the effect of the smoothness parameter and complement the main text with more experiments.

**Weaknesses:**

It would have been nice to touch or at least mention dimensionality reduction techniques and problems with redundant objectives.
Seems its the other side of the coin, given that there is some correlation in the objectives for them to be satisfied or addressed well with a single solution. As shown in Figure 2, each solution address 20 of the 100 objectives, does this mean that the underlying 100 objective problem can be summarized by a 5 objective problem?.

**Questions:**

Is possible this scheme to solve many-objectives problems works well due to some redundancy in the objectives, meaning, objectives are correlated so some solutions that perform well on one of them should also perform well in the positively correlated objectives.
Have you consider some dimensionality reduction techniques to maybe reduce which objectives should be evaluated during the scalarization?

---

> ### Author Response · Authors · 2024-11-24
> **Response to Reviewer CNbj [1/2]**
>
> Thank you very much for your time and effort in reviewing our work. Our responses to your concerns are as follows.
>
> >**W1. Dimensionality Reduction for Many Objective Optimization:** It would have been nice to touch or at least mention dimensionality reduction techniques and problems with redundant objectives. Seems its the other side of the coin, given that there is some correlation in the objectives for them to be satisfied or addressed well with a single solution. As shown in Figure 2, each solution address 20 of the 100 objectives, does this mean that the underlying 100 objective problem can be summarized by a 5 objective problem?
>
> Thank you very much for bringing our attention to the research direction on dimensionality reduction. We fully agree with you that dimensionality reduction (focuses on the objectives) is an important counterpart of our proposed method (focuses on the solutions), and should be discussed in this paper.
>
> **Discussion on Dimensionality Reduction.** First of all, following your suggestion, we have added the below brief introduction for dimensionality reduction in the related work section:
>
> *Dimensionality reduction [1,2,3] is a widely used technique to deal with many-objective optimization problems with potential redundant objectives. By summarizing all objectives by a few representative objectives, these methods can reformulate the originally challenging problem into a simpler problem with much fewer objectives. A detailed discussion with the dimensionality reduction can be found in Appendix D.6 due to the page limit.*
>
> [1] Kalyanmoy Deb and Dhish Kumar Saxena. On Finding Pareto-Optimal Solutions Through Dimensionality Reduction for Certain Large-Dimensional Multi-Objective Optimization Problems. KanGAL Report Number 2005011, 2005.
>
> [2] Dimo Brockhoff and Eckart Zitzler. Are All Objectives Necessary? On Dimensionality Reduction in Evolutionary Multiobjective Optimization. PPSN 2006.
>
> [3] Hemant Kumar Singh, Amitay Isaacs, Tapabrata Ray. A Pareto Corner Search Evolutionary Algorithm and Dimensionality Reduction in Many-Objective Optimization Problems. TEVC 2021.
>
> We also agree with you that the dimensionality reduction technique is very useful for many-objective optimization, especially for problems with highly correlated or even redundant objectives. For example, in Figure 2, we show the result of our proposed method on a synthetic optimization problem with 100 objectives. To clearly demonstrate the behavior of our proposed method, we intentionally construct the problem such that it has $5$ groups of very similar $20$ objectives. Therefore, from the viewpoint of dimensionality reduction, these 20 objectives can be represented by a single objective. In this way, the original 100-objective problem can be summarized by a 5-objective problem as you mention.

---

> ### Author Response · Authors · 2024-11-24
> **Response to Reviewer CNbj [2/2]**
>
> **Viewpoint of Solution Set Optimization.** On the other hand, we also want to show the solution set optimization we investigated in this paper also has it own advantages for many-objective optimization.
>
> - **Requirement from Real-World Application:** The requirement of finding a few solutions to tackle many optimization objectives will naturally arise in many real-word applications such as building a small set of machine learning models (e.g., mixture of experts) to handle many different data or tasks, training a few agents to handle a large number of clients in federated learning, and producing a few different versions of advertisements to serve a large group of diverse audiences. For these cases, it is more natural to tackle the problem from the viewpoint of solution set optimization.
>
> - **No Redundant Objectives:** In many real-world problems, the objectives we care about are not redundant or not highly correlated with each other. By keeping all objectives, the solution set optimization method can more flexibly tackle the trade-offs among all different objectives. For example, when all objectives are conflicting, our proposed (S)TCH-Set will actually find a Pareto solution with the optimal (S)TCH scalarization value for each group of objectives while the groups are adaptively assigned during the optimization process. If the best solution-objective group assignment is known, it is analogous to running the traditional (S)TCH scalarization to find a Pareto solution for each group of objectives. The flexibility of the few-solutions-for-many-objectives approach over the few-solutions-for-few-summarized-objectives is also an interesting research topic in future work.
>
> - **Bridging Different Approaches:** To some degree, our proposed (S)TCH-Set method can be treated as a generalized (smooth) clustering method that assigns each solution to tackle different groups of solutions. If we only keep one representative objective for each group, it could become a dimensionality reduction approach at the end. In addition, if we treat the proposed (S)TCH-Set scalarization function as an indicator, it is actually an indicator-based method for many-objective optimization. We hope this paper can inspire more interesting follow-up works on bridging these different but related approaches.
>
> We have added the above discussion in Appendix D.6 in the revised paper.
>
> > **Q1. Objectives Redundancy for (S)TCH-Set:** Is possible this scheme to solve many-objectives problems works well due to some redundancy in the objectives, meaning, objectives are correlated so some solutions that perform well on one of them should also perform well in the positively correlated objectives. Have you consider some dimensionality reduction techniques to maybe reduce which objectives should be evaluated during the scalarization?
>
> We hope the discussions above can already clarify the relation between dimensionality reduction and our proposed method for solution set optimization. Our proposed method will work well for problems with some redundant objectives, where one solution will be assigned to tackle all the redundant objectives together. For problems without redundant objectives, our proposed (S)TCH-Set scalarization method can still find a set of few solutions to minimize the worst values of all objectives as much as possible.
>
> In this work, we did not use any dimensionality reduction techniques to reduce the objectives during the optimization process. However, it could be a useful approach to significantly reduce the computational overhead, especially when there are redundant objectives in the problem. We leave the investigation of this research direction to future work.

---

> > ### Comment · Reviewer_CNbj · 2024-12-03
> >
> > Something that I missed mentioning in my initial review is that maybe the method STCH-Set could be used to discover if there is some objective redundancy. Ideally one could run this method for a few iterations and by observing how many objectives are covered by a single solution explore if there exist some correlation.
> >
> > There is an error in line 1619, seems the answer to my question was copied as is in the discussion on the appendix.
> >
> > I think the work is very interesting and is a nice extension of an scalarization function that is very successful in multi-objective optimization and evolutionary multi-objective optimization and its smooth version for gradient based optimizers.
> > I support the acceptance of this work.

---

> ### Author Response · Authors · 2024-12-03
>
> Thank you very much for your support and for pointing out the error in the appendix. We will carefully revise the appendix and the whole paper.
>
>
> We very much appreciate you pointing out the connection between STCH-Set and dimensionality reduction and bringing our attention to this promising research direction that we ignored. We will explore more ideas in this direction in future work.

---

### Meta-Review · Area_Chair_HvkY · 2024-12-18

**Metareview:**

This paper studies the problem of finding an approximate set of Pareto solutions in multi-objective optimization. To overcome the curse of dimensionality, the authors propose a novel Tchebycheff set scalarization method to find representative solutions  to cover a large number of objectives. All the reviewers support the acceptance of this paper, and some minor issues have been clarified during the rebuttal period.  Therefore, I recommend "Accept".

**Additional Comments On Reviewer Discussion:**

The discussion is quite fruitful in my mind. Great job!

---

### Decision · Program_Chairs · 2025-01-22

Accept (Poster)